# A Dual-Augmented Block Minimization Framework for Learning with Limited Memory

**Ian E.H. Yen** [*]    **Shan-Wei Lin** [†]    **Shou-De Lin** [†]

[*] University of Texas at Austin          [†] National Taiwan University

[*] `ianyen@cs.utexas.edu`    `{r03922067,sdlin}@csie.ntu.edu.tw`

## Abstract

In past few years, several techniques have been proposed for training of linear Support Vector Machine (SVM) in limited-memory setting, where a dual block-coordinate descent (dual-BCD) method was used to balance cost spent on I/O and computation. In this paper, we consider the more general setting of regularized *Empirical Risk Minimization (ERM)* when data cannot fit into memory. In particular, we generalize the existing block minimization framework based on strong duality and *Augmented Lagrangian* technique to achieve global convergence for general convex ERM. The block minimization framework is flexible in the sense that, given a solver working under sufficient memory, one can integrate it with the framework to obtain a solver globally convergent under limited-memory condition. We conduct experiments on L1-regularized classification and regression problems to corroborate our convergence theory and compare the proposed framework to algorithms adopted from online and distributed settings, which shows superiority of the proposed approach on data of size ten times larger than the memory capacity.

## 1   Introduction

Nowadays data of huge scale are prevalent in many applications of statistical learning and data mining. It has been argued that model performance can be boosted by increasing both number of samples and features, and through crowdsourcing technology, annotated samples of terabytes storage size can be generated [3]. As a result, the performance of model is no longer limited by the sample size but the amount of available computational resources. In other words, the data size can easily go beyond the size of physical memory of available machines. Under this setting, most of learning algorithms become slow due to expensive I/O from secondary storage device [26].

When it comes to huge-scale data, two settings are often considered — online and distributed learning. In the online setting, each sample is processed only once without storage, while in the distributed setting, one has several machines that can jointly fit the data into memory. However, the real cases are often not as extreme as these two — there are usually machines that can fit part of the data, but not all of them. In this setting, an algorithm can only process a block of data at a time. Therefore, balancing the time spent on I/O and computation becomes the key issue [26]. Although one can employ an online-fashioned learning algorithm in this setting, it has been observed that online method requires large number of epoches to achieve comparable performance to batch method, and at each epoch it spends most of time on I/O instead of computation [2, 21, 26]. The situation for online method could become worse for problem of non-smooth, non-strongly convex objective function, where a qualitatively slower convergence of online method is exhibited [15, 16] than that proved for strongly-convex problem like SVM [14].

In the past few years, several algorithms have been proposed to solve large-scale linear Support Vector Machine (SVM) in the limited memory setting [2, 21, 26]. These approaches are based on a dual

Block Coordinate Descent (dual-BCD) algorithim, which decomposes the original problem into a series of block sub-problems, each of them requires only a block of data loaded into memory. The approach was proved linearly convergent to the global optimum, and demonstrated fast convergence empirically. However, the convergence of the algorithm relies on the assumption of a smooth dual problem, which, as we show, does not hold generally for other regularized *Empirical Risk Minimizaton (ERM)* problem. As a result, although the dual-BCD approach can be extended to the more general setting, it is not globally convergent except for a class of problems with L2-regularizer.

In this paper, we first show how to adapt the dual block-coordinate descnet method of [2, 26] to the general setting of regularized *Empirical Risk Mimization (ERM)*, which subsumes most of supervised learning problems ranging from classification, regression to ranking and recommendation. Then we discuss the convergence issue arises when the underlying ERM is not strongly-convex. A *Primal Proximal Point* ( or *Dual Augmented Lagrangian* ) method is then proposed to address this issue, which as we show, results in a block minimization algorithm with global convergence to optimum for convex regularized ERM problems. The framework is flexible in the sense that, given a solver working under sufficient-memory condition, it can be integrated into the block minimization framework to obtain a solver globally convergent under limited-memory condition.

We conduct experiments on L1-regularized classification and regression problems to corroborate our convergence theory, which shows that the proposed simple dual-augmented technique changes the convergence behavior dramatically. We also compare the proposed framework to algorithms adopted from online and distributed settings. In particular, we describe how to adapt a distributed optimization framework — Alternating Direction Method of Multiplier (ADMM) [1] — to the limited-memory setting, and show that, although the adapted algorithm is effective, it is not as efficient as the proposed framework specially designed for limited-memory setting. Note our experiment does not adapt into comparison some recently proposed distributed learning algorithms (CoCoA etc.) [7, 10] that only apply to ERM with L2-regularizer or some other distributed method designed for some specific loss function [19].

## 2 Problem Setup

In this work, we consider the regularized *Empirical Risk Minimization* problem, which given a data set $\mathcal{D} = \{(\Phi_n, \boldsymbol{y}_n)\}_{n=1}^N$, estimates a model through

$$\min_{\boldsymbol{w} \in \mathbb{R}^d, \boldsymbol{\xi}_n \in \mathbb{R}^p} \quad F(\boldsymbol{w}, \boldsymbol{\xi}) = R(\boldsymbol{w}) + \sum_{n=1}^N L_n(\boldsymbol{\xi}_n)$$

$$s.t. \quad \Phi_n \boldsymbol{w} = \boldsymbol{\xi}_n, \ n \in [N] \tag{1}$$

where $\boldsymbol{w} \in \mathbb{R}^d$ is the model parameter to be estimated, $\Phi_n$ is a $p$ by $d$ design matrix that encodes features of the $n$-th data sample, $L_n(\boldsymbol{\xi}_n)$ is a convex loss function that penalizes the discrepancy between ground truth and prediction vector $\boldsymbol{\xi}_n \in \mathbb{R}^p$, and $R(\boldsymbol{w})$ is a convex regularization term penalizing model complexity.

The formulation (1) subsumes a large class of statistical learning problems ranging from classification [27], regression [17], ranking [8], and convex clustering [24]. For example, in classification problem, we have $p = |\mathcal{Y}|$ where $\mathcal{Y}$ consists of the set of all possible labels and $L_n(\boldsymbol{\xi})$ can be defined as the logistic loss $L_n(\boldsymbol{\xi}) = \log(\sum_{k \in \mathcal{Y}} \exp(\xi_k)) - \xi_{y_n}$ as in logistic regression or the hinge loss $L_n(\boldsymbol{\xi}) = \max_{k \in \mathcal{Y}}(1 - \delta_{k,y_n} + \xi_k - \xi_{y_n})$ as used in support vector machine; in a (multi-task) regression problem, the target variable consists of $K$ real values $\mathcal{Y} = \mathbb{R}^K$, the prediction vector has $p = K$ dimensions, and a square loss $L_n(\boldsymbol{\xi}) = \frac{1}{2}\|\boldsymbol{\xi} - \boldsymbol{y}_n\|_2^2$ is often used. There are also a variety of regularizers $R(\boldsymbol{w})$ employed in different applications, which includes the L2-regularizer $R(\boldsymbol{w}) = \frac{\lambda}{2}\|\boldsymbol{w}\|^2$ in ridge regression, L1-regularizer $R(\boldsymbol{w}) = \lambda\|\boldsymbol{w}\|_1$ in *Lasso*, nuclear-norm $R(\boldsymbol{w}) = \lambda\|\boldsymbol{w}\|_*$ in matrix completion, and a family of structured group norms $R(\boldsymbol{w}) = \lambda\|\boldsymbol{w}\|_{\mathcal{G}}$ [11]. Although the specific form of $L_n(\boldsymbol{\xi})$, $R(\boldsymbol{w})$ does not affect the implementation of the limited-memory training procedure, two properties of the functions — strong convexity and smoothness — have key effects on the behavior of the block minimization algorithm.

**Definition 1** (Strong Convexity). *A function $f(x)$ is strongly convex iff it is lower bounded by a simple quadratic function*

$$f(\boldsymbol{y}) \geq f(\boldsymbol{x}) + \nabla f(\boldsymbol{x})^T(\boldsymbol{y} - \boldsymbol{x}) + \frac{m}{2}\|\boldsymbol{x} - \boldsymbol{y}\|^2 \tag{2}$$

*for some constant $m > 0$ and $\forall \boldsymbol{x}, \boldsymbol{y} \in dom(f)$.*

**Definition 2** (Smoothness). *A function $f(x)$ is smooth iff it is upper bounded by a simple quadratic function*

$$f(\boldsymbol{y}) \leq f(\boldsymbol{x}) + \nabla f(\boldsymbol{x})^T(\boldsymbol{y} - \boldsymbol{x}) + \frac{M}{2}\|\boldsymbol{x} - \boldsymbol{y}\|^2 \tag{3}$$

*for some constant $0 \leq M < \infty$ and $\forall \boldsymbol{x}, \boldsymbol{y} \in dom(f)$.*

For instance, the square loss and logistic loss are both smooth and strongly convex [1], while the hinge-loss satisfies neither of them. On the other hand, most of regularizers such as L1-norm, structured group norm, and nuclear norm are neither smooth nor strongly convex, except for the L2-regularizer, which satifies both. In the following we will demonstrate the effects of these properties to Block Minimization algorithms.

Throughout this paper, we will assume that a solver for (1) that works in sufficient-memory condition is given, and our task is to design an algorithmic framework that integrates with the solver to efficiently solve (1) when data cannot fit into memory. We will assume, however, that the $d$-dimensional parameter vector $\boldsymbol{w}$ can be fit into memory.

## 3 Dual Block Minimization

In this section, we extend the block minimization framework of [26] from linear SVM to the general setting of regularized ERM (1).The dual of (1) can be expressed as

$$\min_{\boldsymbol{\mu} \in \mathbb{R}^d, \boldsymbol{\alpha}_n \in \mathbb{R}^p} \quad R^*(-\boldsymbol{\mu}) + \sum_{n=1}^{N} L_n^*(\boldsymbol{\alpha}_n)$$

$$s.t. \quad \sum_{n=1}^{N} \Phi_n^T \boldsymbol{\alpha}_n = \boldsymbol{\mu} \tag{4}$$

where $R^*(-\boldsymbol{\mu})$ is the convex conjugate of $R(\boldsymbol{w})$ and $L_n^*(\boldsymbol{\alpha}_n)$ is the convex conjugate of $L_n(\boldsymbol{\xi}_n)$. The block minimization algorithm of [26] basically performs a dual Block-Coordinate Descent (dual-BCD) over (4) by dividing the whole data set $\mathcal{D}$ into $K$ blocks $\mathcal{D}_{B_1}, ..., \mathcal{D}_{B_K}$, and optimizing a block of dual variables $(\boldsymbol{\alpha}_{B_k}, \boldsymbol{\mu})$ at a time, where $\mathcal{D}_{B_k} = \{(\Phi_n, \boldsymbol{y}_n)\}_{n \in B_k}$ and $\boldsymbol{\alpha}_{B_k} = \{\boldsymbol{\alpha}_n | n \in B_k\}$.

In [26], the dual problem (4) is derived explicitly in order to perform the algorithm. However, for many sparsity-inducing regularizer such as L1-norm and nuclear norm, it is more efficient and convenient to solve (1) in the primal [6, 28]. Therefore, here instead of explicitly forming the dual problem, we express it implicitly as

$$G(\boldsymbol{\alpha}) = \min_{\boldsymbol{w}, \boldsymbol{\xi}} \quad \mathcal{L}(\boldsymbol{\alpha}, \boldsymbol{w}, \boldsymbol{\xi}), \tag{5}$$

where $\mathcal{L}(\boldsymbol{\alpha}, \boldsymbol{w}, \boldsymbol{\xi})$ is the Lagrangian function of (1), and maximize (5) w.r.t. a block of variables $\boldsymbol{\alpha}_{B_k}$ from the primal instead of dual by strong duality

$$\max_{\boldsymbol{\alpha}_{B_k}} \left\{ \min_{\boldsymbol{w}, \boldsymbol{\xi}} \mathcal{L}(\boldsymbol{\alpha}, \boldsymbol{w}, \boldsymbol{\xi}) \right\} = \min_{\boldsymbol{w}, \boldsymbol{\xi}} \left\{ \max_{\boldsymbol{\alpha}_{B_k}} \mathcal{L}(\boldsymbol{\alpha}, \boldsymbol{w}, \boldsymbol{\xi}) \right\} \tag{6}$$

with other dual variables $\{\boldsymbol{\alpha}_{B_j} = \boldsymbol{\alpha}_{B_j}^t\}_{j \neq k}$ fixed. The maximization of dual variables $\boldsymbol{\alpha}_{B_k}$ in (6) then enforces the primal equalities $\Phi_n \boldsymbol{w} = \boldsymbol{\xi}_n$, $n \in B_k$, which results in the block minimization problem

$$\min_{\boldsymbol{w} \in \mathbb{R}^d, \boldsymbol{\xi}_n \in \mathbb{R}^p} \quad R(\boldsymbol{w}) + \sum_{n \in B_k} L_n(\boldsymbol{\xi}_n) + \boldsymbol{\mu}_{B_k}^{tT} \boldsymbol{w}$$

$$s.t. \quad \Phi_n \boldsymbol{w} = \boldsymbol{\xi}_n, \ n \in B_k, \tag{7}$$

where $\boldsymbol{\mu}_{B_k}^t = \sum_{n \notin B_k} \Phi_n^T \boldsymbol{\alpha}_n^t$. Note that, in (7), variables $\{\boldsymbol{\xi}_n\}_{n \notin B_k}$ have been dropped since they are not relevant to the block of dual variables $\boldsymbol{\alpha}_{B_k}$, and thus given the $d$ dimensional vector $\boldsymbol{\mu}_{B_k}^t$, one can solve (7) without accessing data $\{(\Phi_n, \boldsymbol{y}_n)\}_{n \notin B_k}$ outside the block $B_k$. Throughout the dual-BCD algorithm, we maintain $d$-dimensional vector $\boldsymbol{\mu}^t = \sum_{n=1}^{N} \Phi_n^T \boldsymbol{\alpha}_n^t$ and compute $\boldsymbol{\mu}_B^t$ via

$$\boldsymbol{\mu}_B^t = \boldsymbol{\mu}^t - \sum_{n \in B_k} \Phi_n^T \boldsymbol{\alpha}_n^t \qquad (8)$$

in the beginning of solving each block subproblem (7). Since subproblem (7) is of the same form to the original problem (1) except for one additional linear augmented term $\boldsymbol{\mu}_{B_k}^T \boldsymbol{w}$, one can adapt the solver of (1) to solve (7) easily by providing an augmented version of the gradient

$$\nabla_{\boldsymbol{w}} \bar{F}(\boldsymbol{w}, \boldsymbol{\xi}) = \nabla_{\boldsymbol{w}} F(\boldsymbol{w}, \boldsymbol{\xi}) + \boldsymbol{\mu}_{B_k}^t$$

to the solver, where $\bar{F}(.)$ denotes the function with augmented terms and $F(.)$ denotes the function without augmented terms. Note the augmented term $\boldsymbol{\mu}_{B_k}^t$ is constant and separable w.r.t. coordinates, so it adds little overhead to the solver. After obtaining solution $(\boldsymbol{w}^*, \boldsymbol{\xi}_{B_k}^*)$ from (7), we can derive the corresponding optimal dual variables $\boldsymbol{\alpha}_{B_k}$ for (6) according to the KKT condition and maintain $\boldsymbol{\mu}$ subsequently by

$$\boldsymbol{\alpha}_n^{t+1} = \nabla_{\boldsymbol{\xi}_n} L_n(\boldsymbol{\xi}_n^*), \ n \in B_k \qquad (9)$$

$$\boldsymbol{\mu}^{t+1} = \boldsymbol{\mu}_{B_k}^t + \sum_{n \in B_k} \Phi_n^T \boldsymbol{\alpha}_n^{t+1}. \qquad (10)$$

The procedure is summarized in Algorithm 1, which requires a total memory capacity of $O(d + |\mathcal{D}_{B_k}| + p|B_k|)$. The factor $d$ comes from the storage of $\boldsymbol{\mu}^t$, $\boldsymbol{w}^t$, factor $|\mathcal{D}_{B_k}|$ comes from the storage of data block, and the factor $p|B_k|$ comes from the storage of $\boldsymbol{\alpha}_{B_k}$. Note this requires the same space complexity as that required in the original algorithm proposed for linear SVM [26], where $p = 1$ for the binary classification setting.

## 4 Dual-Augmented Block Minimization

The Block Minimization Algorithm 1, though can be applied to the general regularized ERM problem (1), it is not guaranteed that the sequence $\{\boldsymbol{\alpha}^t\}_{t=0}^{\infty}$ produced by Algorithm 1 converges to global optimum of (1). In fact, the global convergence of Algorithm 1 only happens for some special cases. One sufficient condition for the global convergence of a *Block-Coordinate* Descent algorithm is that the terms in objective function that are not separable w.r.t. blocks must be *smooth* (Definition 2).

The dual objective function (4) (expressed using only $\boldsymbol{\alpha}$) comprises two terms $R^*(-\sum_{n=1}^{N} \Phi_n^T \boldsymbol{\alpha}_n) + \sum_{n=1}^{N} L_n^*(\boldsymbol{\alpha}_n)$, where second term is separable w.r.t. to $\{\boldsymbol{\alpha}_n\}_{n=1}^{N}$, and thus is also separable w.r.t. $\{\boldsymbol{\alpha}_{B_k}\}_{k=1}^{K}$, while the first term couples variables $\boldsymbol{\alpha}_{B_1}, ..., \boldsymbol{\alpha}_{B_K}$ involving all the blocks. As a result, if $R^*(-\boldsymbol{\mu})$ is a smooth function according to Definition 2, then Algorithm 1 has global convergence to the optimum. However, the following theorem states this is true only when $R(\boldsymbol{w})$ is strongly convex.

**Theorem 1** (Strong/Smooth Duality). *Assume $f(.)$ is closed and convex. Then $f(.)$ is smooth with parameter $M$ if and only if its convex conjugate $f^*(.)$ is strongly convex with parameter $m = \frac{1}{M}$.*

A proof of above theorem can be found in [9]. According to Theorem 1, the Block Minimization Algorithm 1 is not globally convergent if $R(\boldsymbol{w})$ is not strongly convex, which however, is the case for most of regularizers other than the L2-norm $R(\boldsymbol{w}) = \frac{1}{2}\|\boldsymbol{w}\|^2$, as discussed in Section 2.

In this section, we propose a remedy to this problem, which by a Dual-Augmented Lagrangian method (or equivalently, Primal Proximal Point method), creates a dual objective function of desired property that iteratively approaches the original objective (1), and results in fast global convergence of the dual-BCD approach.

**Algorithm 1** Dual Block Minimization

1. Split data $\mathcal{D}$ into blocks $B_1, B_2, ..., B_K$.
2. Initialize $\boldsymbol{\mu}^0 = \mathbf{0}$.
**for** $t = 0, 1, ...$ **do**
   3.1. Draw $k$ uniformly from $[K]$.
   3.2. Load $\mathcal{D}_{B_k}$ and $\boldsymbol{\alpha}_{B_k}^t$ into memory.
   3.3. Compute $\boldsymbol{\mu}_{B_k}^t$ from (8).
   3.4. Solve (7) to obtain $(\boldsymbol{w}^*, \boldsymbol{\xi}_{B_k}^*)$.
   3.5. Compute $\boldsymbol{\alpha}_{B_k}^{t+1}$ by (9).
   3.6. Maintain $\boldsymbol{\mu}^{t+1}$ through (10).
   3.7. Save $\boldsymbol{\alpha}_{B_k}^{t+1}$ out of memory.
**end for**

**Algorithm 2** Dual-Aug. Block Minimization

1. Split data $\mathcal{D}$ into blocks $B_1, B_2, ..., B_K$.
2. Initialize $\boldsymbol{w}^0 = \mathbf{0}, \boldsymbol{\mu}^0 = \mathbf{0}$.
**for** $t = 0, 1, ...$ (outer iteration) **do**
   **for** $s = 0, 1, ..., S$ **do**
     3.1.1. Draw $k$ uniformly from $[K]$.
     3.1.2. Load $\mathcal{D}_{B_k}, \boldsymbol{\alpha}_{B_k}^s$ into memory.
     3.1.3. Compute $\boldsymbol{\mu}_{B_k}^s$ from (15).
     3.1.4. Solve (14) to obtain $(\boldsymbol{w}^*, \boldsymbol{\xi}_{B_k}^*)$.
     3.1.5. Compute $\boldsymbol{\alpha}_{B_k}^{s+1}$ by (16).
     3.1.6. Maintain $\boldsymbol{\mu}^{s+1}$ through (17).
     3.1.7. Save $\boldsymbol{\alpha}_{B_k}^{s+1}$ out of memory.
   **end for**
   3.2. $\boldsymbol{w}^{t+1} = \boldsymbol{w}^*(\boldsymbol{\alpha}^S)$.
**end for**

## 4.1 Algorithm

The *Dual Augmented Lagrangian (DAL)* method (or equivalently, *Proximal Point Method*) modifies the original problem by introducing a sequence of Proximal Maps

$$\boldsymbol{w}^{t+1} = arg\min_{\boldsymbol{w}} \quad F(\boldsymbol{w}) + \frac{1}{2\eta_t}\|\boldsymbol{w} - \boldsymbol{w}^t\|^2, \tag{11}$$

where $F(\boldsymbol{w})$ denotes the ERM problem (1) Under this simple modification, instead of doing Block-Coordinate Descent in the dual of original problem (1), we perform Dual-BCD on the proximal sub-problem (11). As we show in next section, the dual formulation of (11) has the required property for global convergence of the Dual BCD algorithm — all terms involving more than one block of variables $\boldsymbol{\alpha}_{B_k}$ are smooth. Given the current iterate $\boldsymbol{w}^t$, the Dual-Augmented Block Minimization algorithm optimizes the dual of proximal-point problem (11) w.r.t. one block of variables $\boldsymbol{\alpha}_{B_k}$ at a time, keeping others fixed $\{\boldsymbol{\alpha}_{B_j} = \boldsymbol{\alpha}_{B_j}^{(t,s)}\}_{j \neq k}$:

$$\max_{\boldsymbol{\alpha}_{B_k}} \min_{\boldsymbol{w}, \boldsymbol{\xi}} \mathcal{L}(\boldsymbol{w}, \boldsymbol{\xi}, \boldsymbol{\alpha}) = \min_{\boldsymbol{w}, \boldsymbol{\xi}} \max_{\boldsymbol{\alpha}_{B_k}} \mathcal{L}(\boldsymbol{w}, \boldsymbol{\xi}, \boldsymbol{\alpha}) \tag{12}$$

where $\mathcal{L}(.)$ is the Lagrangian of (11)

$$\mathcal{L}(\boldsymbol{w}, \boldsymbol{\xi}, \boldsymbol{\alpha}) = F(\boldsymbol{w}, \boldsymbol{\xi}) + \sum_{n=1}^{N} \boldsymbol{\alpha}_n^T(\Phi_n \boldsymbol{w} - \boldsymbol{\xi}_n) + \frac{1}{2\eta_t}\|\boldsymbol{w} - \boldsymbol{w}^t\|^2. \tag{13}$$

Once again, the maximization w.r.t. $\boldsymbol{\alpha}_{B_k}$ in (12) enforces the equalities $\Phi_n \boldsymbol{w} = \boldsymbol{\xi}_n$, $n \in B_k$ and thus leads to a primal sub-problem involving only data in block $B_k$:

$$\min_{\boldsymbol{w} \in \mathbb{R}^d, \boldsymbol{\xi}_n \in \mathbb{R}^p} \quad R(\boldsymbol{w}) + \sum_{n \in B_k} L_n(\boldsymbol{\xi}_n) + \boldsymbol{\mu}_{B_k}^{(t,s)T}\boldsymbol{w} + \frac{1}{2\eta_t}\|\boldsymbol{w} - \boldsymbol{w}_t\|^2$$
$$s.t. \quad \Phi_n \boldsymbol{w} = \boldsymbol{\xi}_n, \ n \in B_k, \tag{14}$$

where $\boldsymbol{\mu}_{B_k}^{(t,s)} = \sum_{n \notin B_k} \Phi_n^T \boldsymbol{\alpha}_n^{(t,s)}$. Note that (14) is almost the same as (7) except that it has a proximal-point augmented term. Therefore, one can follow the same procedure as in Algorithm 1 to maintain the vector $\boldsymbol{\mu}^{(t,s)} = \sum_{n=1}^N \Phi_n^T \boldsymbol{\alpha}_n^{(t,s)}$ and computes

$$\boldsymbol{\mu}_{B_k}^{(t,s)} = \boldsymbol{\mu}^{(t,s)} - \sum_{n \in B_k} \Phi_n^T \boldsymbol{\alpha}_n^{(t,s)} \tag{15}$$

before solving each block subproblem (14). After obtaining solution $(\boldsymbol{w}^*, \boldsymbol{\xi}_{B_k}^*)$ from (14), we update dual variables $\boldsymbol{\alpha}_{B_k}$ as

$$\boldsymbol{\alpha}_n^{(t,s+1)} = \nabla_{\boldsymbol{\xi}_n} L_n(\boldsymbol{\xi}_n^*), \ n \in B_k. \tag{16}$$

and maintain $\boldsymbol{\mu}$ subsequently as

$$\boldsymbol{\mu}^{(t,s+1)} = \boldsymbol{\mu}_{B_k}^{(t,s)} + \sum_{n \in B_k} \Phi_n^T \boldsymbol{\alpha}_n^{(t,s+1)}. \tag{17}$$

The sub-problem (14) is of similar form to the original ERM problem (1). Since the augmented term is a simple quadratic function separable w.r.t. each coordinate, given a solver for (1) working in sufficient-memory condition, one can easily adapt it by modifying

$$\nabla_{\boldsymbol{w}} \bar{F}(\boldsymbol{w}, \boldsymbol{\xi}) = \nabla_{\boldsymbol{w}} F(\boldsymbol{w}, \boldsymbol{\xi}) + \boldsymbol{\mu}_{B_k}^t + (\boldsymbol{w} - \boldsymbol{w}_t)/\eta_t$$
$$\nabla_{\boldsymbol{w}}^2 \bar{F}(\boldsymbol{w}, \boldsymbol{\xi}) = \nabla_{\boldsymbol{w}}^2 F(\boldsymbol{w}, \boldsymbol{\xi}) + I/\eta_t,$$

where $\bar{F}(.)$ denotes the function with augmented terms and $F(.)$ denotes the function without augmented terms. The Block Minimization procedure is repeated until every sub-problem (14) reaches a tolerance $\epsilon_{in}$. Then the proximal point method update $\boldsymbol{w}^{t+1} = \boldsymbol{w}^*(\boldsymbol{\alpha}^{(t,s)})$ is performed, where $\boldsymbol{w}^*(\boldsymbol{\alpha}^{(t,s)})$ is the solution of (14) for the latest dual iterate $\boldsymbol{\alpha}^{(t,s)}$. The resulting algorithm is summarized in Algorithm 2.

## 4.2 Analysis

In this section, we analyze the convergence rate of Algorithm 2 to the optimum of (1). First, we show that the proximal-point formulation (11) has a dual problem with desired property for the global convergence of Block-Coordinate Descent. In particular, since the dual of (11) takes the form

$$\min_{\boldsymbol{\alpha}_n \in \mathbb{R}^p} \quad \tilde{R}^*\left(-\sum_{n=1}^{N} \Phi_n^T \boldsymbol{\alpha}_n\right) + \sum_{n=1}^{N} L_n^*(\boldsymbol{\alpha}_n) \tag{18}$$

where $\tilde{R}^*(.)$ is the convex conjugate of $\tilde{R}(\boldsymbol{w}) = R(\boldsymbol{w}) + \frac{1}{2\eta_t}\|\boldsymbol{w} - \boldsymbol{w}_t\|^2$, and since $\tilde{R}(\boldsymbol{w})$ is strongly convex with parameter $m = 1/\eta_t$, the convex conjugate $\tilde{R}^*(.)$ is smooth with parameter $M = \eta_t$ according to Theorem 1. Therefore, (18) is in the composite form of a convex, smooth function plus a convex, block-separable function. This type of function has been widely studied in the literature of Block-Coordinate Descent [13]. In particular, one can show that a Block-Coordinate Descent applied on (18) has global convergence to optimum with a fast rate by the following theorem.

**Theorem 2** (BCD Convergence). *Let the sequence $\{\boldsymbol{\alpha}^s\}_{s=1}^{\infty}$ be the iterates produced by Block Coordinate Descent in the inner loop of Algorithm 2, and $K$ be the number of blocks. Denote $\tilde{F}^*(\boldsymbol{\alpha})$ as the dual objective function of (18) and $\tilde{F}_{opt}^*$ the optimal value of (18). Then with probability $1-\rho$,*

$$\tilde{F}^*(\boldsymbol{\alpha}^s) - \tilde{F}_{opt}^* \leq \epsilon, \quad for \quad s \geq \beta K \log\left(\frac{\tilde{F}^*(\boldsymbol{\alpha}^0) - \tilde{F}_{opt}^*}{\rho\epsilon}\right) \tag{19}$$

*for some constant $\beta > 0$ if (i) $L_n(.)$ is smooth, or (ii) $L_n(.)$ is polyhedral function and $R(.)$ is also polyhedral or smooth. Otherwise, for any convex $L_n(.), R(.)$ we have*

$$\tilde{F}^*(\boldsymbol{\alpha}^s) - \tilde{F}_{opt}^* \leq \epsilon, \quad for \quad s \geq \frac{cK}{\epsilon} \log\left(\frac{\tilde{F}^*(\boldsymbol{\alpha}^0) - \tilde{F}_{opt}^*}{\rho\epsilon}\right) \tag{20}$$

*for some constant $c > 0$.*

Note the above analysis (in appendix) does not assume exact solution of each block subproblem. Instead, it only assumes each block minimization step leads to a dual ascent amount proportional to that produced by a single (dual) proximal gradient ascent step on the block of dual variables. For the outer loop of Primal Proximal-Point (or Dual Augmented Lagrangian) iterates (11), we show the following convergence theorem.

**Theorem 3** (Proximal Point Convergence). *Let $F(\boldsymbol{w})$ be objective of the regularized ERM problem (1), and $\mathcal{R} = \max_{\boldsymbol{v}} \max_{\boldsymbol{w}} \{\|\boldsymbol{v} - \boldsymbol{w}\| : F(\boldsymbol{w}) \leq F(\boldsymbol{w}^0), F(\boldsymbol{v}) \leq F(\boldsymbol{w}^0)\}$ be the radius of initial level set. The sequence $\{\boldsymbol{w}^t\}_{t=1}^{\infty}$ produced by the Proximal-Point update (11) with $\eta_t = \eta$ has*

$$F(\boldsymbol{w}^{t+1}) - F_{opt} \leq \epsilon, \quad for \quad t \geq \tau \log\left(\frac{\omega}{\epsilon}\right). \tag{21}$$

*for some constant $\tau, \omega > 0$ if both $L_n(.)$ and $R(.)$ are (i) strictly convex and smooth or (ii) polyhedral. Otherwise, for any convex $F(\boldsymbol{w})$ we have*

$$F(\boldsymbol{w}^{t+1}) - F_{opt} \leq \mathcal{R}^2/(2\eta t).$$

The following theorem further shows that solving sub-problem (11) inexactly with tolerance $\epsilon/t$ suffices for convergence to $\epsilon$ overall precision, where $t$ is the number of outer iterations required.

**Theorem 4** (Inexact Proximal Map). *Suppose, for a given dual iterate $\boldsymbol{w}^t$, each sub-problem (11) is solved inexactly s.t. the solution $\hat{\boldsymbol{w}}^{t+1}$ has*

$$\|\hat{\boldsymbol{w}}^{t+1} - \mathbf{prox}_{\eta_t F}(\boldsymbol{w}^t)\| \leq \epsilon_0. \tag{22}$$

*Then let $\{\hat{\boldsymbol{w}}^t\}_{t=1}^{\infty}$ be the sequence of iterates produced by inexact proximal updates and $\{\boldsymbol{w}^t\}_{t=1}^{\infty}$ as that generated by exact updates. After $t$ iterations, we have*

$$\|\hat{\boldsymbol{w}}^t - \boldsymbol{w}^t\| \leq t\epsilon_0. \tag{23}$$

Note for $L_n(.)$, $R(.)$ being strictly convex and smooth, or polyhedral, $t$ is of order $O(\log(1/\epsilon))$, and thus it only requires $O(K \log(1/\epsilon) \log(t/\epsilon)) = O(K \log^2(1/\epsilon))$ overall number of block minimization steps to achieve $\epsilon$ suboptimality. Otherwise, as long as $L_n(.)$ is smooth, for any convex regularizer $R(.)$, $t$ is of order $O(1/\epsilon)$, so it requires $O(K(1/\epsilon) \log(t/\epsilon)) = O(\frac{K \log(1/\epsilon)}{\epsilon})$ total number of block minimization steps.

## 4.3 Practical Issues

### 4.3.1 Solving Sub-Problem Inexactly

While the analysis in Section 4.2 assumes exact solution of subproblems, in practice, the Block Minimization framework does not require solving subproblem (11), (14) exactly. In our experiments, it suffices for the fast convergence of proximal-point update (11) to solve subproblem (14) for only a single pass of all blocks of variables $\boldsymbol{\alpha}_{B_1},..., \boldsymbol{\alpha}_{B_K}$, and limit the number of iterations the designated solver spends on each subproblem (7), (14) to be no more than some parameter $T_{max}$.

### 4.3.2 Random Selection w/o Replacement

In Algorithm 1 and 2, the block to be optimized is chosen uniformly at random from $k \in \{1, ..., K\}$, which eases the analysis for proving a better convergence rate [13]. However, in practice, to avoid unbalanced update frequency among blocks, we do *random sampling without replacement* for both Algorithm 1 and 2, that is, for every $K$ iterations, we generate a random permutation $\pi_1, ..., \pi_K$ of block index $1, .., K$ and optimize block subproblems (7), (14) according to the order $\pi_1, .., \pi_K$. This also eases the checking of inner-loop stopping condition.

### 4.3.3 Storage of Dual Variables

Both the algorithms 1 and 2 need to store the dual variables $\boldsymbol{\alpha}_{B_k}$ into memory and load/save them from/to some secondary storage units, which requires a time linear to $p|B_k|$. For some problems, such as multi-label classification with large number of labels or structured prediction with large number of factors, this can be very expensive. In this situation, one can instead maintain $\boldsymbol{\mu}_{\bar{B}_k} = \sum_{n \in B_k} \Phi_n^T \alpha_n = \boldsymbol{\mu} - \boldsymbol{\mu}_{B_k}$ directly. Note $\boldsymbol{\mu}_{\bar{B}_k}$ has I/O and storage cost linear to $d$, which can be much smaller than $p|B_k|$ in a low-dimensional problem.

## 5 Experiment

In this section, we compare the proposed *Dual Augmented Block Minimization* framework (Algorithm 2) to the vanilla Dual Block Coordinate Descent algorithm [26] and methods adopted from Online and Distributed Learning. The experiments are conducted on the problem of $L1$-regularized L2-loss SVM [27] and the (*Lasso*) (L1-regularized Regression) problem [17] in the limited-memory setting with data size 10 times larger than the available memory. For both problems, we use state-of-the-art randomized coordinate descent method [13, 27] as the solver for solving sub-problems (7), (14), (59), (63), and we set parameter $\eta_t = 1$, $\lambda = 1$ (of L1-regularizer) for all experiments. Four public benchmark data sets are used— *webspam*, *rcv1-binary* for classification and *year-pred*, *E2006* for regression, which can be obtained from the LIBSVM data set collections. For *year-pred* and *E2006*, the features are generated from Random Fourier Features [12, 23] that approximate the effect of Gaussian RBF kernel. Table 1 summarizes the data statistics. The algorithms in comparison and their shorthands are listed below, where all solvers are implemented in C/C++ and run on 64-bit machine with 2.83GHz Intel(R) Xeon(R) CPU. We constrained the process to use no more than $1/10$ of memory required to store the whole data.

- **OnlineMD**: Stochastic Mirror Descent method specially designed for L1-regularized problem proposed in [15] with step size chosen from $10^{-2}$-$10^2$ for best performance.

Table 1: Data Statistics: Summary of data statistics when stored using sparse format. The last two columns specify memory consumption in (MB) of the whole data and that of a block when data is split into $K = 10$ partitions.

| Data | #train | #test | dimension | #non-zeros | Memory | Block |
|---|---|---|---|---|---|---|
| webspam | 315,000 | 31,500 | 680,714 | 1,174,704,031 | 20,679 | 2,068 |
| rcv1 | 202,420 | 20,242 | 7,951,176 | 656,977,694 | 12,009 | 1,201 |
| year-pred | 463,715 | 51,630 | 2,000 | 927,893,715 | 13,702 | 1,370 |
| E2006 | 16,087 | 3,308 | 30,000 | 8,088,636 | 8,088 | 809 |

Figure 1: Relative function value difference to the optimum and Testing RMSE (Accuracy) on LASSO (top) and L1-regularized L2-SVM (bottom). (RMSE best for year-pred: 9.1320; for E2006: 0.4430), (Accuracy best for for webspam: 0.4761%; best for rcv1: 2.213%).

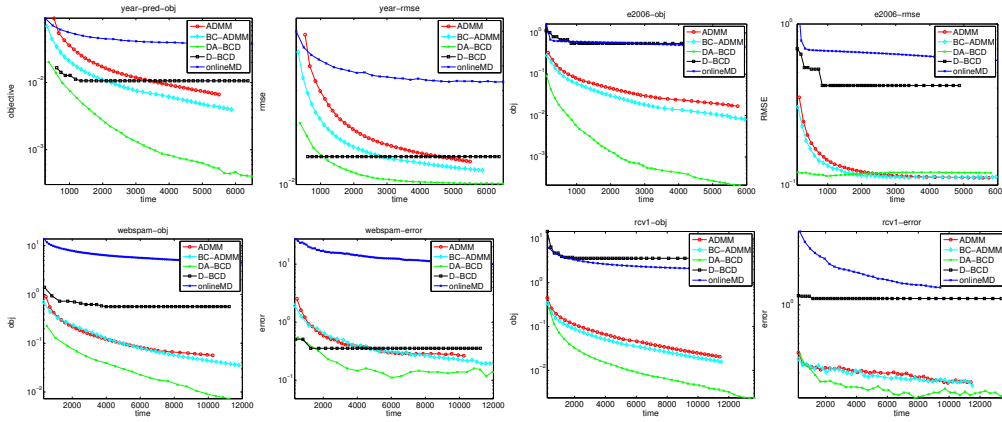

- **D-BCD**[2]: Dual Block-Coordinate Descent method (Algorithm 1).
- **DA-BCD**: Dual-Augmented Block Minimization (Algorithm 2).
- **ADMM**: ADMM for limited-memory learning (Algorithm 3 in appendix-B).
- **BC-ADMM**: Block-Coordinate ADMM that updates a randomly chosen block of dual variables at a time for limited-memory learning (Algorithm 4 in appendix-B) .

We use *wall clock time* that includes both I/O and computation as measure for training time in all experiments. In Figure 5, three measures are plotted versus the training time: Relative objective function difference to the optimum, Testing RMSE and Accuracy. Figure 5 shows the results, where as expected, the dual Block Coordinate Descent (D-BCD) method without augmentation cannot improve the objective after certain number of iterations. However, with extremely simple modification, the Dual-Augmented Block Minimization (DA-BCD) algorithm becomes not only globally convergent but with a rate several times faster than other approaches. Among all methods, the convergence of *Online Mirror Descent* (SMIDAS) is significantly slower, which is expected since (i) the online Mirror Descent on a non-smooth, non-strongly convex function converges at a rate qualitatively slower than the linear convergence rate of DA-BCD and ADMM [15, 16], and (ii) Online method does not utilize the available memory capacity and thus spends unbalanced time on I/O and computation. For methods adopted from distributed optimization, the experiment shows BC-ADMM consistently, but only slightly, improves ADMM, and both of them converge much slower than the DA-BCD approach, presumably due to the conservative updates on the dual variables.

**Acknowledgement** We thank to the support of Telecommunication Lab., Chunghwa Telecom Co., Ltd via TL-103-8201, AOARD via No. FA2386-13-1-4045, Ministry of Science and Technology, National Taiwan University and Intel Co. via MOST102-2911-I-002-001, NTU103R7501, 102-2923-E-002-007-MY2, 102-2221-E-002-170, 103-2221-E-002-104-MY2.

## Footnotes

[1]The logistic loss is strongly convex when its input $\boldsymbol{\xi}$ are within a bounded range, which is true as long as we have a non-zero regularizer $R(\boldsymbol{w})$.

[2]The objective value obtained from D-BCD fluctuates a lot, in figures we plot the lowest values achieved by D-BCD from the beginning to time $t$.

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
