[Supplementary Material 1 · LimitedMemNIPS_final_full.10-20.pdf]

# 6 Appendix-A. Convergence Analysis

## 6.1 Convergence of Randomized Block Coordinate Descent

We first establish the linear convergence of Randomized Block Coordinate Descent (RBCD) when $L_n(.)$ is smooth in the sense that its first derivative $L'_n(.)$ is Lipschitz-continuous with parameter $M_L$, which then implies $L_n^*(\boldsymbol{\alpha}_n)$ is strongly convex with parameter $1/M_L$.

**Theorem 2-1** (Dual-RBCD for Smooth Loss). *Let the sequence $\{\boldsymbol{\alpha}^s\}_{s=1}^\infty$ be the iterates produced by RBCD in the inner loop of Algorithm 2, and $K$ be the number of blocks. Denote $\tilde{F}^*(\boldsymbol{\alpha})$ as the dual objective function of (18) and $\tilde{F}_{opt}^*$ the optimal function value of (18). Then with probability $1 - \rho$,*

$$\tilde{F}^*(\boldsymbol{\alpha}^s) - \tilde{F}_{opt}^* \le \epsilon, \ \ for \ \ s \ge \frac{K}{1 - c_1} \log(\frac{\tilde{F}^*(\boldsymbol{\alpha}^0) - \tilde{F}_{opt}^*}{\rho\epsilon}) \tag{24}$$

*if $L_n(.)$ is smooth, where $0 < c_1 < 1$ is a constant depends on the smoothness parameter of $L_n(.)$.*

*Proof.* This is a special case of theorem 6 and theorem 4 in [13], where they consider composite objective function of the form

$$F(\boldsymbol{\alpha}) = f(\boldsymbol{\alpha}) + \Psi(\boldsymbol{\alpha}), \tag{25}$$

where $f(\boldsymbol{\alpha})$ is a convex, smooth function, and $\Psi(\boldsymbol{\alpha})$ is a convex, block-separable function. In our case,

$$f(\boldsymbol{\alpha}) = \tilde{R}^*(-\sum_{n=1}^N \Phi_n^T \boldsymbol{\alpha}_n), \ \Psi(\boldsymbol{\alpha}) = \sum_{n=1}^N L_n^*(\boldsymbol{\alpha}_n). \tag{26}$$

Note $\tilde{R}^*(.)$ is smooth w.r.t. $\boldsymbol{\alpha}_{B_k}$ with parameter $M_R = \eta_t B^2$, where $B \ge \|\Phi_{B_k}\|_2$ is an upper bound on the $\ell_2$-norm of each block's design matrix. If the loss function $L_n(.)$ is smooth with paramter $M_L$, by Theorem 1, $\Psi(\boldsymbol{\alpha})$ is strongly convex with parameter $1/M_L$, and thus, based on [theorem 6, 21], (24) holds with

$$c_1 = \begin{cases} 1 - \frac{1}{4M_R M_L} & , \ if M_R M_L \ge \frac{1}{2} \\ M_R M_L & , \ o.w.. \end{cases} \tag{27}$$

$\square$

For some important classes of ERM, such as Support Vector Machine (SVM) and its variants (e.g. Multiclass, Structral SVM), $L_n(\boldsymbol{\alpha}_n)$ is not smooth but piecewise-linear. In the following, we show that the linear convergence of RBCD holds for any loss $L_n(\boldsymbol{\alpha}_n)$ with polyhedral epigraph if $R(\boldsymbol{w})$ is also polyhedral or smooth. The proof utilizes a restricted version of Strong Convexity called Constant Nullspace Strong Convexity [20, 22] and obtains a much tighter bound for RBCD than the bound proved in [20] for general feasible descent method. The proof follows is a generalization of that in [25] for proving linear convergence of RCD applied to the Augmented Lagrangian of Linear Program.

The augmented dual objective function (25), after some algebraic rearrangement, is equivalent to

$$\min_{\boldsymbol{\alpha}, \boldsymbol{\mu}} \ \ \sum_{n=1}^N L_n^*(\boldsymbol{\alpha}_n) + R^*(-\boldsymbol{\mu}) + \frac{\eta_t}{2} \|\sum_{n=1}^N \Phi_n^T \boldsymbol{\alpha}_n - \boldsymbol{\mu} + \boldsymbol{w}^t/\eta_t\|^2 \tag{28}$$

up to a constant. For $L_n^*(\boldsymbol{\alpha}_n)$, $R^*(-\boldsymbol{\mu})$ being polyhedral, their epigraphs $\mathbf{epi}(L_n)$, $\mathbf{epi}(R)$ are polyhedrons and thus (28) can be also formulated as

$$\begin{aligned} \min_{\boldsymbol{\alpha}, \boldsymbol{\mu}, \boldsymbol{t}, s} & \ \ \sum_{n=1}^N t_n + r + \frac{\eta_t}{2} \|\sum_{n=1}^N \Phi_n^T \boldsymbol{\alpha}_n - \boldsymbol{\mu} + \boldsymbol{w}^t/\eta_t\|^2 \\ s.t. & \ \ (\boldsymbol{\alpha}_n, t_n) \in \mathbf{epi}(L_n) \\ & \ \ (\boldsymbol{\mu}, r) \in \mathbf{epi}(R), \end{aligned} \tag{29}$$

which is of the form

$$\begin{aligned} \min_{\boldsymbol{x}} & \ \ F(\boldsymbol{x}) = g(\bar{\Phi}^T \boldsymbol{x}) + \boldsymbol{c}^T \boldsymbol{x} \\ s.t. & \ \ \boldsymbol{x} \in \mathcal{P} \end{aligned} \tag{30}$$

where $g(\boldsymbol{z}) = \frac{\eta_t}{2}\|\boldsymbol{z} + \boldsymbol{w}^t/\eta_t\|^2$ is a strongly convex function, $\mathcal{P}$ is a polyhedral set
$$\mathcal{P} = \{(\boldsymbol{\alpha}, \boldsymbol{t}, \boldsymbol{\mu}, r) \mid (\boldsymbol{\alpha}_n, t_n) \in \mathbf{epi}(L_n), (\boldsymbol{\mu}, r) \in \mathbf{epi}(R)\},$$

and

$$\boldsymbol{x} = \begin{bmatrix} \boldsymbol{\alpha} \\ \boldsymbol{t} \\ \boldsymbol{\mu} \\ r \end{bmatrix} \qquad \bar{\Phi} = \begin{bmatrix} \Phi \\ O_{N,d} \\ -I_{d,d} \\ O_{1,d} \end{bmatrix} \qquad \boldsymbol{c} = \begin{bmatrix} \mathbf{0} \\ \mathbf{1} \\ \mathbf{0} \\ 1 \end{bmatrix}.$$

We will use $I_{\boldsymbol{\alpha}}$, $I_{\boldsymbol{t}}$, $I_{\boldsymbol{\mu}}$ and $I_s$ denote the set of variable indexes $j$ that correspond to $\boldsymbol{\alpha}$, $\boldsymbol{t}$, $\boldsymbol{\mu}$ and $r$ respectively. For this type of objective function, we can show that the set of optimal solutions is a polyhedron defined by the following Lemma.

**Lemma 1** (Lemma 4.2 of [20]). *The optimal solutions to problem* (30) *forms a polyhedral set*
$$\mathcal{S} = \{\boldsymbol{x} \mid \bar{\Phi}^T \boldsymbol{x} = \boldsymbol{p}^*, \ \boldsymbol{c}^T \boldsymbol{x} = q^*, \ \boldsymbol{x} \in \mathcal{P}\} \tag{31}$$
*for some unique $\boldsymbol{p}^*$, $q^*$.*

Furthermore, we can utilize the Hoffman's bound (defined in the following) to bound the distance of any point $\boldsymbol{x}$ to the optimal solution set $\mathcal{S}$.

**Lemma 2** (Hoffman's Bound). *Let $\mathcal{S} = \{\boldsymbol{x} \in \mathbb{R}^d \mid A\boldsymbol{x} \leq \boldsymbol{c}, \ E\boldsymbol{x} = \boldsymbol{c}\}$ be a polyhedral set. Then for any point $\boldsymbol{x} \in \mathbb{R}^d$,*
$$\|\boldsymbol{x} - \Pi_{\mathcal{S}}(\boldsymbol{x})\|_2^2 \leq \theta \left\| \begin{bmatrix} [A\boldsymbol{x} - \boldsymbol{c}]_+ \\ E\boldsymbol{x} - \boldsymbol{c} \end{bmatrix} \right\|_2^2 \tag{32}$$
*where $\Pi_{\mathcal{S}}(\boldsymbol{x}) = arg\min_{\boldsymbol{y} \in \mathcal{S}} \|\boldsymbol{y} - \boldsymbol{x}\|$ is the projection of $\boldsymbol{x}$ to the set $\mathcal{S}$, and $\theta > 0$ is a constant depending on the polyhedral set $\mathcal{S}$.*

*Proof.* The Hoffman's bound first appears in [4] and a proof for the $\ell_2$-norm's version (32) and the definition of the constant $\theta(\mathcal{S})$ can be found in [20] (lemma 4.3). $\qquad\square$

By Lemma 2, for any feasible $\boldsymbol{x} \in \mathcal{P}$, we obtain error bound
$$\|\boldsymbol{x} - \boldsymbol{x}^*\|^2 \leq \theta(\mathcal{S}) \left( \|\bar{\Phi}^T \boldsymbol{x} - \boldsymbol{p}^*\|^2 + \|\boldsymbol{c}^T \boldsymbol{x} - q^*\|^2 \right), \tag{33}$$
which plays a crucial role in the proof of linear convergence.

The RBCD algorithm performed on (25) can be considered as minimizing (29) w.r.t. a block of dual variables $\{(\boldsymbol{\alpha}_n, t_n)\}_{n \in B_k}$ together with $(\boldsymbol{\mu}, s)$, while fixing all other variables $\{(\boldsymbol{\alpha}_n, t_n)\}_{n \notin B_k}$. In the following, we show that each block minimization step leads to a significant progress.

**Lemma 3** (Descent Amount). *The expected descent amount for each Block Minimization step of Algorithm 2 has*
$$\mathbb{E}[F(\boldsymbol{x}^{k+1})] - F(\boldsymbol{x}^k) \leq \frac{1}{K} \left( \min_{\boldsymbol{\delta}} \ h(\boldsymbol{x}^k + \boldsymbol{\delta}) + \langle \nabla F(\boldsymbol{x}^k), \boldsymbol{\delta} \rangle + \frac{M\eta_t}{2} \|\boldsymbol{\delta}\|^2 \right), \tag{34}$$

*where*
$$h(\boldsymbol{x}) = \begin{cases} 0, & \boldsymbol{x} \in \mathcal{P} \\ \infty, & o.w. \end{cases} \tag{35}$$
*and $M \geq \max_{k \in [K]} \|\Phi_{B_k}\|_2^2$ denotes a bound on the spectral norm of each block's design matrix.*

*Proof.* First, notice that RBCD optimizes the function form of only variable $\boldsymbol{\alpha}$ while maintains other variables $(\boldsymbol{t}, \boldsymbol{\mu}, s)$ as their optimal values in each block minimization step, so we have
$$\mathbf{0} = \min_{\boldsymbol{\mu}, r} \ h(\boldsymbol{x}) + \nabla F(\boldsymbol{x}^s)^T(\boldsymbol{x} - \boldsymbol{x}^s) + \frac{M\eta_t}{2}\|\boldsymbol{x} - \boldsymbol{x}^s\|^2. \tag{36}$$
The algorithm picks coordinate uniformly from $\{(\boldsymbol{\alpha}_{B_k}, \boldsymbol{t}_{B_k})\}_{k=1}^K$ to update. Since the constant $M$ upper bounds $\|\nabla_{\boldsymbol{\alpha}_{B_k}, \boldsymbol{t}_{B_k}} F(\boldsymbol{x})\|_2^2$, we have
$$\begin{aligned} F(\boldsymbol{x}^{s+1}) - F(\boldsymbol{x}^s) &= F(\boldsymbol{\alpha}^{s+1}, t^*(\boldsymbol{\alpha}^{s+1}), \boldsymbol{\mu}^*(\boldsymbol{\alpha}^{s+1}), r^*(\boldsymbol{\alpha}^{s+1})) - F(\boldsymbol{x}^s) \\ &\leq F(\boldsymbol{\alpha}^{s+1}, t^*(\boldsymbol{\alpha}^{s+1}), \boldsymbol{\mu}^s, r^s) - F(\boldsymbol{x}^s) \\ &\leq \min_{\boldsymbol{\delta}_{B_k}} \ h(\boldsymbol{x}^s + \boldsymbol{\delta}_{B_k}) + \nabla_{B_k} F(x^k)^T \boldsymbol{\delta}_{B_k} + \frac{M\eta_t}{2}\|\boldsymbol{\delta}_{B_k}\|^2. \end{aligned}$$

where $\boldsymbol{\delta}_{B_k}$ denotes a change of variables restricted on $(\Delta \boldsymbol{\alpha}_{B_k}, \Delta \boldsymbol{t}_{B_k})$ with all other variables fixed. Note the minimization in (69) is seperable w.r.t $\{\boldsymbol{\delta}_{B_k}\}_{k=1}^K$. Therefore, taking expectation of LHS and RHS w.r.t. $k$ yields the result. $\qquad \square$

Before moving on, note that function $g(\boldsymbol{z}) = \frac{\eta_t}{2} \|\boldsymbol{z} + \boldsymbol{w}^t/\eta_t\|^2$ is locally Lipschitz-continuous with constant $L_g = \eta_t R_z$ for $\boldsymbol{z}$ satisfying $\|\boldsymbol{z} + \boldsymbol{w}^t/\eta_t\| \le R_z$, that is,

$$|g(\boldsymbol{z}_1) - g(\boldsymbol{z}_2)| \le L_g \|\boldsymbol{z}_1 - \boldsymbol{z}_2\| \tag{37}$$

for $\forall \boldsymbol{z}_1, \boldsymbol{z}_2$ with $\|\boldsymbol{z}_1 + \boldsymbol{w}^t/\eta_t\| \le R_z, \|\boldsymbol{z}_2 + \boldsymbol{w}^t/\eta_t\| \le R_z$, where $R_z$ is an upper bound on the magnitude of iterates $\|\boldsymbol{w}^{t+1}\|/\eta_t = \|\bar{\Phi}^T \boldsymbol{x}^t + \boldsymbol{w}^t/\eta_t\|$.

From simplicity of analysis, in the following, we slightly loosen upper bounds by setting constants $L_g \leftarrow \max(L_g, 1)$, $M \leftarrow \max(M, 1)$, $\theta \leftarrow \max(\theta, 1)$, such that $L_g, M, \theta \ge 1$. Then we are ready to prove the main theorem of this section.

**Theorem 5** (Linear Convergence). *The iterates $\{\boldsymbol{x}^s\}_{s=0}^{\infty}$ of Block Minimization for polyhedral $L_n(.)$, $R(.)$ satisfy*

$$\mathbb{E}[F(\boldsymbol{x}^{s+1})] - F^* \le \left(1 - \frac{1}{K\gamma}\right)(\mathbb{E}[F(\boldsymbol{x}^s)] - F^*)$$

*where $F^*$ is the optimum of* (28) *and*

$$\gamma = \max\left\{16\eta_t M\theta(F^0 - F^*), \ 2M\theta(1 + 4L_g^2), \ 6\right\}.$$

*Proof.* Let $\boldsymbol{x}^* = \Pi_{\mathcal{S}}(\boldsymbol{x}^s)$ be the projection of $\boldsymbol{x}^s$ to the set of optimal solutions. From Lemma 3, we have

$$
\begin{aligned}
\mathbb{E}[F(\boldsymbol{x}^{s+1})] - F(\boldsymbol{x}^s) &\le \frac{1}{K}\left(\min_{\boldsymbol{\delta}} \ h(\boldsymbol{x}^s + \boldsymbol{\delta}) + \langle \nabla F(\boldsymbol{x}^s), \boldsymbol{\delta}\rangle + \frac{M\eta_t}{2}\|\boldsymbol{\delta}\|^2\right) \\
&\le \frac{1}{K}\left(\min_{\boldsymbol{\delta}} \ h(\boldsymbol{x}^s + \boldsymbol{\delta}) + F(\boldsymbol{x}^s + \boldsymbol{\delta}) - F(\boldsymbol{x}^s) + \frac{M\eta_t}{2}\|\boldsymbol{\delta}\|^2\right) \\
&\le \frac{1}{K}\left(\min_{a\in[0,1]} \ F(\boldsymbol{x}^s + a(\boldsymbol{x}^* - \boldsymbol{x}^s)) - F(\boldsymbol{x}^s) + \frac{M\eta_t a^2}{2}\|\boldsymbol{x}^* - \boldsymbol{x}^s\|^2\right) \\
&\le \frac{1}{K}\left(\min_{a\in[0,1]} \ -a(F(\boldsymbol{x}^s) - F(\boldsymbol{x}^*)) + \frac{M\eta_t a^2}{2}\|\boldsymbol{x}^* - \boldsymbol{x}^s\|^2\right),
\end{aligned}
\tag{38}
$$

where the second and fourth inequality follow from the convexity of $F(\boldsymbol{x})$, and the third inequality follows from the fact that both $\boldsymbol{x}^*$ and $\boldsymbol{x}^s$ are feasible ($h(\boldsymbol{x}^*) = h(\boldsymbol{x}^s) = 0$). Now based on the error bound inequality (68), we discuss two cases.

**Case 1:** $4L_g^2\|\bar{\Phi}^T\boldsymbol{x} - \boldsymbol{p}^*\|^2 < (\boldsymbol{c}^T\boldsymbol{x} - q^*)^2$.

In this case, we have

$$
\begin{aligned}
\|\boldsymbol{x}^s - \boldsymbol{x}^*\|^2 &\le \theta\left(\|\bar{\Phi}^T\boldsymbol{x}^s - \boldsymbol{p}^*\|^2 + \|\boldsymbol{c}^T\boldsymbol{x}^s - q^*\|^2\right) \\
&\le \theta\left(\frac{1}{4L_g^2} + 1\right)(\boldsymbol{c}^T\boldsymbol{x}^s - q^*)^2 \le 2\theta(\boldsymbol{c}^T\boldsymbol{x}^s - q^*)^2
\end{aligned}
\tag{39}
$$

and

$$|\boldsymbol{c}^T\boldsymbol{x}^s - q^*| \ge 2L_g\|\bar{\Phi}^T\boldsymbol{x}^s - \boldsymbol{p}^*\| \ge 2|g(\bar{\Phi}^T\boldsymbol{x}^s) - g(\boldsymbol{p}^*)|.$$

Note in this case, $\boldsymbol{c}^T\boldsymbol{x}^s - q^*$ must be non-negative. Otherwise,

$$
\begin{aligned}
F(\boldsymbol{x}^s) - F^* &= g(\bar{\Phi}^T\boldsymbol{x}^s) - g(\boldsymbol{p}^*) + (\boldsymbol{c}^T\boldsymbol{x}^s - q^*) \\
&\le |g(\bar{\Phi}^T\boldsymbol{x}^s) - g(\boldsymbol{p}^*)| - |\boldsymbol{c}^T\boldsymbol{x}^s - q^*| \\
&\le -\frac{1}{2}|\boldsymbol{c}^T\boldsymbol{x}^s - q^*| < 0,
\end{aligned}
$$

leads to contradiction (since $\boldsymbol{x}^s$ is feasible, $F(\boldsymbol{x}^s)$ cannot be smaller than $F^*$). Therefore, we have

$$
\begin{aligned}
F(\boldsymbol{x}^s) - F^* &= g(\bar{\Phi}^T \boldsymbol{x}^s) - g(\boldsymbol{p}^*) + \boldsymbol{c}^T \boldsymbol{x}^s - q^* \\
&\geq -|g(\bar{\Phi}^T \boldsymbol{x}^s) - g(\boldsymbol{p}^*)| + \boldsymbol{c}^T \boldsymbol{x}^s - q^* \\
&\geq \frac{1}{2}(\boldsymbol{c}^T \boldsymbol{x}^s - q^*).
\end{aligned} \tag{40}
$$

Combining (38), (39), and (40), we have

$$
\begin{aligned}
\mathbb{E}[F(\boldsymbol{x}^{s+1})] - F(\boldsymbol{x}^s) &\leq \frac{1}{K} \min_{a \in [0,1]} -\frac{a}{2}(\boldsymbol{c}^T \boldsymbol{x}^s - q^*) + \frac{2\eta_t M \theta a^2}{2}(\boldsymbol{c}^T \boldsymbol{x}^s - q^*)^2 \\
&= \begin{cases} -1/(16\eta_t M \theta K) &, \ 1/(4\eta_t M \theta (\boldsymbol{c}^T \boldsymbol{x}^s - q^*)) \leq 1 \\ -\frac{1}{4K}(\boldsymbol{c}^T \boldsymbol{x}^s - q^*) &, \ o.w. \end{cases}
\end{aligned}
$$

Furthermore, we have

$$
-\frac{1}{16\eta_t M \theta K} \leq -\frac{1}{16\eta_t M \theta K (F^0 - F^*)} \left( F(\boldsymbol{x}^*) - F^* \right)
$$

where $F^0 = F(\boldsymbol{x}^0)$, and

$$
-\frac{1}{4K}(\boldsymbol{c}^T \boldsymbol{x}^s - q^*) \leq -\frac{1}{6K}(F(\boldsymbol{x}^s) - F^*)
$$

since $F(\boldsymbol{x}^s) - F^* \leq |g(\bar{\Phi}^T \boldsymbol{x}^s) - g(\boldsymbol{p}^*)| + \boldsymbol{c}^T \boldsymbol{x}^s - q^* \leq \frac{3}{2}(\boldsymbol{c}^T \boldsymbol{x}^s - q^*)$. In summary, for Case 1 we obtain

$$
\mathbb{E}[F(\boldsymbol{x}^{s+1})] - F^* \leq (1 - \frac{1}{K\gamma_1}) \left( \mathbb{E}[F(\boldsymbol{x}^s)] - F^* \right) \tag{41}
$$

where

$$
\gamma_1 = \max \left\{ 16\eta_t M \theta (F^0 - F^*), \ 6 \right\}. \tag{42}
$$

**Case 2:** $4L_g^2 \|\bar{\Phi}^T \boldsymbol{x}^s - \boldsymbol{p}^*\|^2 \geq (\boldsymbol{c}^T \boldsymbol{x}^s - q^*)^2$.

In this case, we have

$$
\|\boldsymbol{x}^s - \boldsymbol{x}^*\|^2 \leq \theta \left( 1 + 4L_g^2 \right) \|\bar{\Phi}^T \boldsymbol{x}^s - \boldsymbol{p}^*\|^2, \tag{43}
$$

and by strong convexity of $g(\boldsymbol{z})$,

$$
F(\boldsymbol{x}^s) - F^* \geq \boldsymbol{c}^T (\boldsymbol{x}^s - \boldsymbol{x}^*) + \nabla g(\boldsymbol{p}^*)^T \bar{\Phi}^T (\boldsymbol{x}^s - \boldsymbol{x}^*) + \frac{\eta_t}{2} \|\bar{\Phi}^T \boldsymbol{x}^s - \boldsymbol{p}^*\|^2.
$$

Adding inequality $0 = h(\boldsymbol{x}^s) - h(\boldsymbol{x}^*) \geq \langle \boldsymbol{\rho}^*, \boldsymbol{x}^s - \boldsymbol{x}^* \rangle$ for some $\boldsymbol{\rho}^* \in \partial h(\boldsymbol{x}^*)$ to the above gives

$$
F(\boldsymbol{x}^s) - F^* \geq \frac{\eta_t}{2} \|\bar{\Phi}^T \boldsymbol{x}^s - \boldsymbol{p}^*\|^2 \tag{44}
$$

since $\boldsymbol{\rho}^* + \boldsymbol{c} + \nabla g(\boldsymbol{p}^*)^T \bar{\Phi}^T = \boldsymbol{\rho}^* + \nabla F(\boldsymbol{x}^*) = 0$. Combining (38), (43), and (44), we obtain

$$
\begin{aligned}
\mathbb{E}[F(\boldsymbol{x}^{s+1})] - F(\boldsymbol{x}^s) &\leq \frac{1}{K} \min_{a \in [0,1]} -a(F(\boldsymbol{x}^s) - F^*) + \frac{M \theta (1 + 4L_g^2) a^2}{2} (F(\boldsymbol{x}^s) - F^*) \\
&= -\frac{1}{2M\theta(1 + 4L_g^2)K} (F(\boldsymbol{x}^s) - F^*)
\end{aligned} \tag{45}
$$

Combining results of Case 1 (41) and Case 2 (45), and taking expectation on both sides w.r.t. the history leads to the result. $\qquad\square$

**Theorem 2-2** (Dual-RBCD for Polyhedral Loss). *Let the sequence $\{\boldsymbol{\alpha}^s\}_{s=1}^{\infty}$ be the iterates produced by RBCD in the inner loop of Algorithm 2, and $K$ be the number of blocks. Denote $\tilde{F}^*(\boldsymbol{\alpha})$ as the augmented dual objective function (18) and $\tilde{F}_{opt}^*$ the optimum of (18). With probability $1 - \rho$,*

$$
\tilde{F}^*(\boldsymbol{\alpha}^s) - \tilde{F}_{opt}^* \leq \epsilon, \ \text{for} \ s \geq \gamma K \log(\frac{\tilde{F}^*(\boldsymbol{\alpha}^0) - \tilde{F}_{opt}^*}{\rho \epsilon}) \tag{46}
$$

*for some constant $\gamma$ if $L_n(.)$ and $R(.)$ are polyhedral.*

*Proof.* This simply applies Theorem 1 of [13] to transfer the linear convergence in expectation into high-probability iteration complexity. $\qquad\square$

## 6.2 Convergence of Proximal-Point Method

The proof of Theorem 3 comprises two parts. The first part proves linear convergence of Proximal-Point update under assumption that both loss $L_n(.)$ and regularizer $R(.)$ are either strictly convex and smooth or polyhedral. The second part proves a sublinear-type convergence depending on parameter $\eta$ that holds for general convex function. The second part can be found in, for example, Theorem 2 of [18]. Here we prove the first part.

Here we prove linear convergence of ALM on problem (25) by leveraging some lemmas provided in the recent advance of analysis for Alternating Direction Method of Multiplier (ADMM) [5]. In particular, by taking Proximal-Point updates (or, equivalently, the ALM updates) as performing gradient descent on the convex, smooth function

$$G(\tilde{\boldsymbol{w}}) = \min_{\boldsymbol{w}} \sum_n L_n(\bar{\Phi}_n \boldsymbol{w}) + R(\boldsymbol{w}) + \frac{1}{2\eta}\|\boldsymbol{w} - \tilde{\boldsymbol{w}}\|^2 \tag{47}$$

and utilizing error bound proved in [5], we show that the Proximal-Point method linearly converges to the optimum of objective (25).

The following lemma establishes the fact $G(\tilde{\boldsymbol{w}})$ is smooth and its gradient $\nabla G(\tilde{\boldsymbol{w}})$ is Lipschitz continuous with modulus $\frac{1}{\eta}$.

**Lemma 4.** *The gradient of $G(\tilde{\boldsymbol{w}})$ is of the form*

$$\nabla G(\tilde{\boldsymbol{w}}) = -(\sum_{n=1}^{N} \Phi_n^T \boldsymbol{\alpha}_n(\tilde{\boldsymbol{w}}) - \boldsymbol{\mu}(\tilde{\boldsymbol{w}})) \tag{48}$$

*where $\boldsymbol{\alpha}_n(\tilde{\boldsymbol{w}})$, $\boldsymbol{\mu}(\tilde{\boldsymbol{w}})$ are minimizers of (28). Furthermore, the gradient $\nabla G(\tilde{\boldsymbol{w}})$ is Lipschitz continuous with modulus $\frac{1}{\eta}$.*

*Proof.* The convex objective function (25) fits the form of objective investigated in Multi-block ADMM [5]. Therefore, the theorem follows directly from Lemma 2.1, 2.2 of [5] respectively. $\square$

As a result of Lemma 4, the proximal-point update is exactly gradient descent of step size $\eta$, which when performed on a smooth function $G(\tilde{\boldsymbol{w}})$, guarantees descent amount

$$G(\boldsymbol{w}^{t+1}) - G(\boldsymbol{w}^t) \leq -\frac{\eta\|\nabla G(\boldsymbol{w}^t)\|^2}{2}. \tag{49}$$

The following theorem then guarantees linear convergence of ALM on our objective (25).

**Theorem 6.** *Denote $S$ as the set of optimal solutions to (47) and $\Pi_S(\boldsymbol{w})$ as the projection of $\boldsymbol{w}$ to $S$, and let $G^*$ be the optimal function value. The iterates $\{\boldsymbol{w}^t\}_{t=1}^{\infty}$ produced by proximal-point method have*

$$\|\sum_{n=1}^{N} \Phi_n^T \boldsymbol{\alpha}_n(\tilde{\boldsymbol{w}}) - \boldsymbol{\mu}(\tilde{\boldsymbol{w}})\| = \|\nabla G(\boldsymbol{w}^t)\| \leq \epsilon$$

*for number of iterations*

$$t \geq \frac{4\tau}{\eta} \log(\sqrt{\frac{2(G(\boldsymbol{w}^0) - G^*)}{\eta}} \frac{1}{\epsilon}),$$

*where $\tau > 0$ is a constant depending on $S$ and initial distance to optimal set $\|\boldsymbol{w}^0 - \Pi_S(\boldsymbol{w}^0)\|$.*

*Proof.* Since $L_n(.)$ and $R(.)$ are either strictly convex and smooth or polyhedral, $L_n^*(.)$ and $R^*(.)$ are also strictly convex and smooth or polyhedral. Therefore, problem (25) satisfies Assumption A(a)-A(e) of [5], and thus the error bound

$$G(\tilde{\boldsymbol{w}}) - G^* \leq \tau\|\nabla G(\tilde{\boldsymbol{w}})\|^2 \tag{50}$$

in Lemma 3.1 of [5] applies to $G(\tilde{\boldsymbol{w}})$ with compact domain $\tilde{\boldsymbol{w}} \in R(\boldsymbol{w}^0)$, where $\tau > 0$ is a constant that depends on geometry of $S$ and the initial distance to the set of optimal solutions, and

$$R(\boldsymbol{w}^0) = \left\{\tilde{\boldsymbol{w}} \mid \|\tilde{\boldsymbol{w}} - \Pi_S(\tilde{\boldsymbol{w}})\| \leq \|\boldsymbol{w}^0 - \Pi_S(\boldsymbol{w}^0)\|\right\}.$$

is the set of $\tilde{\boldsymbol{w}}$ that lie within a radius of $\|\boldsymbol{w}^0 - \Pi_S(\boldsymbol{w}^0)\|$ to the set $S$. Note the iterates $\{\boldsymbol{w}^t\}_{t=0}^\infty$ all lie in the set $R(\boldsymbol{w}^0)$ by the non-expansiveness of proximal operation. Therefore, the error bound (50) applies to all iterates. Combining (69) and (50), we have

$$G(\boldsymbol{w}^{t+1}) - G(\boldsymbol{w}^t) \leq -\frac{\eta(G(\boldsymbol{w}^t) - G^*)}{2\tau},$$

which implies linear convergence. Let $\Delta G_t = G(\boldsymbol{w}^t) - G^*$, and we have

$$\Delta G_t \leq (1 - \frac{\eta}{2\tau})^t \Delta G_0 \leq e^{-\frac{\eta t}{2\tau}} \Delta G_0 \leq \epsilon_1$$

when

$$t \geq \frac{2\tau}{\eta} \log(\frac{\Delta G_0}{\epsilon_1}).$$

Furthermore, by smoothness of $\nabla G(.)$, we have

$$\Delta G_t \geq \frac{\eta \|\nabla G(\boldsymbol{w}^t)\|^2}{2}.$$

Therefore, to guarantee $\|\nabla G(\boldsymbol{w}^t)\| \leq \epsilon_2$, it suffices to have

$$\Delta G^t \leq \eta \epsilon_2^2/2,$$

which can be guaranteed by running

$$t \geq \frac{4\tau}{\eta} \log(\sqrt{\frac{2\Delta G_0}{\eta}} \frac{1}{\epsilon_2})$$

iterations. $\qquad\square$

**Theorem 7** (Inexact Proximal Map). *Suppose, for a given dual iterate* $\boldsymbol{w}^t$, *each sub-problem* (11) *is solved inexactly s.t. the solution* $\hat{\boldsymbol{w}}^{t+1}$ *has*

$$\|\hat{\boldsymbol{w}}^{t+1} - \mathbf{prox}_{\eta_t F}(\boldsymbol{w}^t)\| \leq \epsilon_0. \tag{51}$$

*Then let* $\{\hat{\boldsymbol{w}}^t\}_{t=1}^\infty$ *be the sequence of iterates produced by inexact proximal updates and* $\{\boldsymbol{w}^t\}_{t=1}^\infty$ *as that generated by exact updates. After* $t$ *iterations, we have*

$$\|\hat{\boldsymbol{w}}^t - \boldsymbol{w}^t\| \leq t\epsilon_0. \tag{52}$$

*Proof.* By the non-expansiveness of proximal operation,

$$\begin{aligned}
\|\hat{\boldsymbol{w}}^{t+1} - \boldsymbol{w}^{t+1}\| &\leq \|\hat{\boldsymbol{w}}^{t+1} - \mathbf{prox}_{\eta_t F}(\hat{\boldsymbol{w}}^t)\| + \|\mathbf{prox}_{\eta_t F}(\hat{\boldsymbol{w}}^t) - \boldsymbol{w}^{t+1}\| \\
&\leq \epsilon_0 + \|\mathbf{prox}_{\eta_t F}(\hat{\boldsymbol{w}}^t) - \mathbf{prox}_{\eta_t F}(\boldsymbol{w}^t)\| \\
&\leq \epsilon_0 + \|\hat{\boldsymbol{w}}^t - \boldsymbol{w}^t\|.
\end{aligned}$$

Recursively applying the above inequality leads to the conclusion. $\qquad\square$

# 7 Appendix-B. ADMM under Limited Memory

In this section, we show how an algorithm for *distributed optimization* can be adapted for *limited-memory* learning, which then serves as a baseline to methods specially designed for limited-memory environment. In particular, the adaption sequentializes parallel computation performed on multiple machines into a series of tasks performed on single machine, where states and data partition of each simulated machine are loaded from (saved to) secondary storage units beforehand (afterward). As an example, we show how to adapt *Alternating Direction Method of Multiplier (ADMM)*, a recently proposed distributed optimization framework [1], into our setting.

Given a problem of the form

$$\min_{\boldsymbol{w} \in \mathbb{R}^d} \quad \sum_{i=1}^N f_i(\boldsymbol{w}), \tag{53}$$

**Algorithm 3** ADMM (limited memory)

---
1. Split data $\mathcal{D}$ into blocks $B_1, B_2, ..., B_K$.
2. Initialize $\boldsymbol{w}_k^0 = \boldsymbol{0}$, $\boldsymbol{z}^0 = \boldsymbol{0}$, $\boldsymbol{\mu}_k^0 = \boldsymbol{0}$.
**for** $t = 0, 1, ...$ (outer iteration) **do**
   3. $\boldsymbol{z}^{t+1} = \boldsymbol{0}$
  **for** $k = 1, 2, ..., K$ **do**
    4.1. Swap data block $B_k$, $\boldsymbol{w}_k$, $\boldsymbol{\mu}_k$ into memory.
    4.2. $\boldsymbol{w}_k^{t+1} = argmin_{\boldsymbol{w}} \ \mathcal{L}_k(\boldsymbol{w}, \boldsymbol{z}^t, \boldsymbol{\mu}_k^t)$
    4.3. $\boldsymbol{z}^{t+1} + = (\boldsymbol{w}_k^{t+1} + \boldsymbol{\mu}_k^t/\rho)/K$
  **end for**
  **for** $k = 1, 2, ..., K$ **do**
    5.1. Swap $\boldsymbol{w}_k^{t+1}$, $\boldsymbol{\mu}_k^t$ into memory.
    5.2. $\boldsymbol{\mu}_k^{t+1} = \boldsymbol{\mu}_k^t + \eta(\boldsymbol{w}_k^{t+1} - \boldsymbol{z}^{t+1})$.
  **end for**
**end for**

---

**Algorithm 4** Block-Coordinate ADMM (BC-ADMM)

---
1. Split data $\mathcal{D}$ into blocks $B_1, B_2, ..., B_K$.
2. Initialize $\boldsymbol{w}_k^0 = \boldsymbol{0}$, $\boldsymbol{z}^0 = \boldsymbol{0}$, $\boldsymbol{\mu}_k^0 = \boldsymbol{0}$.
**for** $t = 0, 1, ...$ **do**
   3.1. Randomly chosen $k \in \{1..K\}$ w/o replacement.
   3.2. Swap data block $B_k$, $\boldsymbol{w}_k^t$, $\boldsymbol{\mu}_k^t$ into memory.
   3.3. $\boldsymbol{\mu}_k^{t+1} = \boldsymbol{\mu}_k^t + \eta(\boldsymbol{w}_k^t - \boldsymbol{z}^t)$.
   3.4. $\boldsymbol{w}_k^{t+1} = argmin_{\boldsymbol{w}} \ \mathcal{L}_k(\boldsymbol{w}, \boldsymbol{z}^t, \boldsymbol{\mu}_k^{t+1})$
   3.5. $\boldsymbol{z}^{t+1} = \boldsymbol{z}^t + (\boldsymbol{w}_k^{t+1} + \boldsymbol{\mu}_k^{t+1}/\rho)/K - (\boldsymbol{w}_k^t + \boldsymbol{\mu}_k^t/\rho)/K$
**end for**

---

the ADMM framework splits (53) into $K$ smaller sub-problems defined on different data blocks $B_1, B_2, ..., B_K$, and formulate the dual problem of

$$
\begin{aligned}
\min_{\boldsymbol{w}_k, \boldsymbol{z}} \quad & \sum_{k=1}^K f_k(\boldsymbol{w}_k) + \frac{\rho}{2}(\boldsymbol{w}_k - \boldsymbol{z})^2 \\
s.t. \quad & \boldsymbol{w}_k - \boldsymbol{z} = 0, \ k = 1, .., K \ ,
\end{aligned}
\tag{54}
$$

where $f_k(\boldsymbol{w}_k) = \sum_{i \in B_k} f_i(\boldsymbol{w}_k)$, $\boldsymbol{z}$ is the *consensus parameters*, and $\rho > 0$ is a hyper-parameter. The ADMM procedure finds the saddle point of Lagrangian

$$
\max_{\boldsymbol{\mu}_k} \ \min_{\boldsymbol{w}_k, \boldsymbol{z}} \ \mathcal{L}(\boldsymbol{w}, \boldsymbol{z}, \boldsymbol{\mu}) = \sum_{k=1}^K f(\boldsymbol{w}_k) + \boldsymbol{\mu}_k^T(\boldsymbol{w}_k - \boldsymbol{z}) + \frac{\rho}{2}\|\boldsymbol{w}_k - \boldsymbol{z}\|^2
\tag{55}
$$

via the following iterate

$$
\boldsymbol{w}^{t+1} = \underset{\boldsymbol{w}}{argmin} \ \mathcal{L}(\boldsymbol{w}, \boldsymbol{z}^t, \boldsymbol{\mu}^t)
\tag{56}
$$

$$
\boldsymbol{z}^{t+1} = \underset{\boldsymbol{z}}{argmin} \ \mathcal{L}(\boldsymbol{w}^{t+1}, \boldsymbol{z}, \boldsymbol{\mu}^t)
\tag{57}
$$

$$
\boldsymbol{\mu}_k^{t+1} = \boldsymbol{\mu}_k^t + \eta(\boldsymbol{w}_k^{t+1} - \boldsymbol{z}^{t+1}), \ k = 1, ..., K,
\tag{58}
$$

where $\eta$ is a constant step size. Since given $\boldsymbol{z}^t$, $\mathcal{L}(\boldsymbol{w}, \boldsymbol{z}^t, \boldsymbol{\mu}^t)$ is separable w.r.t. $\boldsymbol{w}_1, \boldsymbol{w}_2, ..., \boldsymbol{w}_K$, step (56) can be solved separately for each $\boldsymbol{w}_k$ as

$$
\boldsymbol{w}_k^{t+1} = \underset{\boldsymbol{w}_k}{argmin} \ \mathcal{L}_k(\boldsymbol{w}_k, \boldsymbol{z}^t, \boldsymbol{\mu}_k^t), \ k = 1, .., K.
\tag{59}
$$

Since the bottleneck of iterate lies in (59), ADMM is inherently suitable for distributed optimization via solving the $K$ subproblems (59) on $K$ machines. The only step requiring communication is (57),

which has close-form solution

$$z^{t+1} = \frac{1}{K} \sum_{k=1}^{K} w_k^{t+1} + \mu_k^t / \rho, \tag{60}$$

that is, a simple average over parameters and multipliers. In limited-memory environment, however, only one block $B_k$ of samples can be fit into memory at a time, and thus $K$ times of swapping is required at each iteration. A naive implementation is depicted in Algorithm 3. Note, in some high-dimensional problem, the model parameters $w_k$ and $\mu_k$ can be of comparable size to the data block, and thus need to be stored out of memory. One drawback of algorithm 3 is that the *consensus parameter $z$* is not updated until $K$ subproblems are solved. In Algorithm 4, we propose another adaption that updates the dual variables of one randomly chosen block $B_k$ at a time as follows

$$\mu_k^t = \mu_k^{t-1} + \eta(w_k^t - z^t) \tag{61}$$

$$w_k^{t+1} = \underset{w_k}{argmin} \ \mathcal{L}_k(w_k, z^t, \mu_k^t) \tag{62}$$

$$z^{t+1} = \underset{z}{argmin} \ \mathcal{L}(w_1^t, ..., w_k^{t+1}, ..., w_K^t, z, \mu_k^t). \tag{63}$$

In this version of limited-memory ADMM, the information learnt from one block can be passed to the next subproblem immediately, and consensus parameters $\mu_k$, $w_k$ only need to be swapped once for each iteration. It has been shown that standard ADMM iterates in Algorithm 3 have global linear convergence to the optimum [5]. The following theorem shows the same type of convergence guarantee also applies to Algorithm 4 .

**Theorem 8** (BC-ADMM Convergence). *Consider a regularized ERM problem* (53) *of the form*

$$f_i(w) = L_i(\Phi_i w) + \frac{1}{N} R(w).$$

*Let $d(\mu)$ be the dual function value of problem* (54). *If the loss function $L_i(.)$ is smooth, and one of $L_i(.)$ or $R(.)$ is strongly convex, Algorithm 4 converges to the optimum of* (53) *at a linear rate, that is,*

$$E[\Delta_p^t + \Delta_d^t] \leq \frac{1}{1+\lambda}(\Delta_p^{t-1} + \Delta_d^{t-1}) \tag{64}$$

*for some constant $\lambda > 0$, where*

$$\begin{aligned} \Delta_p^t &= \mathcal{L}(w^{t+1}, z^{t+1}, \mu^t) - d(\mu^t) \\ \Delta_d^t &= d^* - d(\mu^t) \end{aligned} \tag{65}$$

*are the primal and dual residuals at iterate $t$ respectively.*

Though being effective, the adapted algorithm takes little advantage of the sequential nature of limiter-memory setting. In particular, since the distributed learning algorithm is designed to allow parallel updates, the information passed among parallel sub-problems is limited and the updates on dual variables (58), (62) are conservative with step size $\eta$ compared to the exact block-coordinate minimization (12) in the Dual-Augmented Block Minimization framework. Note ADMM can be seen as an approximate Gradient Descent method on the dual, while analysis in coordinate descent literature [13] shows that Block-Coordinate descent can be up to $K$ times faster than Gradient Descent in the worst-conditioned case, where $K$ is the number of blocks.

## 8 Appendix-C. Convergence of Block-Coordinate ADMM

Let $d(\mu) = \min_{w,z} \mathcal{L}(w, z, \mu)$ be the dual objective for $\mu$ and $d^* = \max_\mu d(\mu)$ be the optimal dual objective value, we define primal residual $\Delta_p^t$ and dual residual $\Delta_d^t$ of current iterate $(w^t, z^t, \mu^t)$ as

$$\begin{aligned} \Delta_p^t &= \mathcal{L}(w^{t+1}, z^{t+1}, \mu^t) - d(\mu^t) \\ \Delta_d^t &= d^* - d(\mu^t). \end{aligned} \tag{66}$$

Note $\Delta_p^t \geq 0$, $\Delta_d^t \geq 0$, and $\Delta_d^t = \Delta_p^t = 0$ if only if $(w^t, z^t, \mu^t)$ is optimal.

**Lemma 5** (Dual Iterate). *For all $t \geq 1$,*
$$\Delta_d^t - \Delta_d^{t-1} \leq -\eta(\boldsymbol{w}_k^t - \boldsymbol{z}^t)^T(\bar{\boldsymbol{w}}_k^t - \bar{\boldsymbol{z}}^t),$$
*where $(\bar{\boldsymbol{w}}^t, \bar{\boldsymbol{z}}^t)$ is the solution to $\min_{\boldsymbol{w},\boldsymbol{z}} \mathcal{L}(\boldsymbol{w}, \boldsymbol{z}, \boldsymbol{\mu}^t)$ that is closest to $(\boldsymbol{w}^t, \boldsymbol{z}^t)$.*

*Proof.*
$$
\begin{aligned}
\Delta_d^t - \Delta_d^{t-1} &= d(\boldsymbol{\mu}^{t-1}) - d(\boldsymbol{\mu}^t) \\
&= \mathcal{L}(\bar{\boldsymbol{w}}^{t-1}, \bar{\boldsymbol{z}}^{t-1}, \boldsymbol{\mu}^{t-1}) - \mathcal{L}(\bar{\boldsymbol{w}}^t, \bar{\boldsymbol{z}}^t, \boldsymbol{\mu}^t) \\
&\leq \mathcal{L}(\bar{\boldsymbol{w}}^t, \bar{\boldsymbol{z}}^t, \boldsymbol{\mu}^{t-1}) - \mathcal{L}(\bar{\boldsymbol{w}}^t, \bar{\boldsymbol{z}}^t, \boldsymbol{\mu}^t) \\
&= (\boldsymbol{\mu}^{t-1} - \boldsymbol{\mu}^t)^T(\bar{\boldsymbol{w}}^t - \bar{\boldsymbol{z}}^t) \\
&= (\boldsymbol{\mu}_k^{t-1} - \boldsymbol{\mu}_k^t)^T(\bar{\boldsymbol{w}}_k^t - \bar{\boldsymbol{z}}^t) \\
&= -\eta(\boldsymbol{w}_k^t - \boldsymbol{z}^t)^T(\bar{\boldsymbol{w}}_k^t - \bar{\boldsymbol{z}}^t),
\end{aligned}
$$
where the third inequality follows from definition $(\bar{\boldsymbol{w}}^{t-1}, \bar{\boldsymbol{z}}^{t-1}) = argmin_{\boldsymbol{w},\boldsymbol{z}} \mathcal{L}(\boldsymbol{w}, \boldsymbol{z}, \boldsymbol{\mu}^{t-1})$. $\square$

**Lemma 6** (Primal Iterate). *For all $t \geq 1$,*
$$
\begin{aligned}
\Delta_p^t - \Delta_p^{t-1} &\leq -\rho \left( \|\boldsymbol{w}_k^{t+1} - \boldsymbol{w}_k^t\|^2 + \|\boldsymbol{z}^{t+1} - \boldsymbol{z}^t\|^2 \right) \\
&\quad + \eta \left( \|\boldsymbol{w}_k^t - \boldsymbol{z}^t\|^2 - (\boldsymbol{w}_k^t - \boldsymbol{z}^t)^T(\bar{\boldsymbol{w}}_k^t - \bar{\boldsymbol{z}}^t) \right)
\end{aligned}
$$

*Proof.*
$$
\begin{aligned}
\Delta_p^t - \Delta_p^{t-1} &= \\
\left( \mathcal{L}(\boldsymbol{w}^{t+1}, \boldsymbol{z}^{t+1}, \boldsymbol{\mu}^t) - d(\boldsymbol{\mu}^t) \right) &- \left( \mathcal{L}(\boldsymbol{w}^t, \boldsymbol{z}^t, \boldsymbol{\mu}^{t-1}) - d(\boldsymbol{\mu}^{t-1}) \right),
\end{aligned}
$$
where $d(\boldsymbol{\mu}^{t-1}) - d(\boldsymbol{\mu}^t)$ can be obtained via Lemma 2.1 as
$$d(\boldsymbol{\mu}^{t-1}) - d(\boldsymbol{\mu}^t) = -\eta(\boldsymbol{w}_k^t - \boldsymbol{z}^t)^T(\bar{\boldsymbol{w}}_k^t - \bar{\boldsymbol{z}}^t). \tag{67}$$
It remains to find
$$
\begin{aligned}
\mathcal{L}(\boldsymbol{w}^{t+1}, \boldsymbol{z}^{t+1}, \boldsymbol{\mu}^t) &- \mathcal{L}(\boldsymbol{w}^t, \boldsymbol{z}^t, \boldsymbol{\mu}^{t-1}) = \\
\mathcal{L}(\boldsymbol{w}^{t+1}, \boldsymbol{z}^{t+1}, \boldsymbol{\mu}^t) &- \mathcal{L}(\boldsymbol{w}^t, \boldsymbol{z}^t, \boldsymbol{\mu}^t) + \mathcal{L}(\boldsymbol{w}^t, \boldsymbol{z}^t, \boldsymbol{\mu}^t) - \mathcal{L}(\boldsymbol{w}^t, \boldsymbol{z}^t, \boldsymbol{\mu}^{r-1}).
\end{aligned}
$$
From strong convexity of the augmented term $\frac{\rho}{2}\|\boldsymbol{w}_k - \boldsymbol{z}\|^2$, and that $\boldsymbol{w}^{t+1}, \boldsymbol{z}^{t+1}$ are minimizers for (56) and (57) respectively, we can bound the primal descent amount by
$$
\begin{aligned}
\mathcal{L}(\boldsymbol{w}^{t+1}, \boldsymbol{z}^{t+1}, \boldsymbol{\mu}^t) &- \mathcal{L}(\boldsymbol{w}^t, \boldsymbol{z}^t, \boldsymbol{\mu}^t) \\
&= \mathcal{L}_k(\boldsymbol{w}_k^{t+1}, \boldsymbol{z}^{t+1}, \boldsymbol{\mu}_k^t) - \mathcal{L}_k(\boldsymbol{w}_k^t, \boldsymbol{z}^t, \boldsymbol{\mu}_k^t) \\
&\leq -\rho \left( \|\boldsymbol{w}_k^{t+1} - \boldsymbol{w}_k^t\|^2 + \|\boldsymbol{z}^{t+1} - \boldsymbol{z}^t\|^2 \right).
\end{aligned}
$$
It is also known that
$$\mathcal{L}(\boldsymbol{w}^t, \boldsymbol{z}^t, \boldsymbol{\mu}^t) - \mathcal{L}(\boldsymbol{w}^t, \boldsymbol{z}^t, \boldsymbol{\mu}^{r-1}) = \eta\|\boldsymbol{w}_k^t - \boldsymbol{z}^t\|^2.$$
Therefore,
$$
\begin{aligned}
\mathcal{L}(\boldsymbol{w}^{t+1}, \boldsymbol{z}^{t+1}, \boldsymbol{\mu}^t) &- \mathcal{L}(\boldsymbol{w}^t, \boldsymbol{z}^t, \boldsymbol{\mu}^{t-1}) \\
&\leq -\rho \left( \|\boldsymbol{w}_k^{t+1} - \boldsymbol{w}_k^t\|^2 + \|\boldsymbol{z}^{t+1} - \boldsymbol{z}^t\|^2 \right) + \eta\|\boldsymbol{w}_k^t - \boldsymbol{z}^t\|^2.
\end{aligned}
$$
Combine above inequality with (67), we obtain the conclusion. $\square$

The following theorem guarantees descent of the primal-dual residual $\Delta_p^t + \Delta_d^t$ in expectation for each iteration of BC-ADMM.

**Theorem 9** (Guaranteed Descent). *For step-size $\eta$ sufficiently small,*
$$E[\Delta_p^t + \Delta_d^t] < (\Delta_p^{t-1} + \Delta_d^{t-1})$$
*for all $t \geq 1$, where $E[.]$ is expectation over blocks $k_1, k_2, ..., k_R$ drawn at iteration $t$.*

*Proof.* Define
$$\Delta z_k^t = (w_k^{t+1} + \mu_k^t/\rho)/K - (w_k^t + \mu_k^{t-1}/\rho)/K$$
and
$$\Delta z^t = \frac{1}{K}\sum_{k=1}^{K}(w_k^{t+1} + \mu_k^t/\rho) - \frac{1}{K}\sum_{k=1}^{K}(w_k^t + \mu_k^{t-1}/\rho)$$

By Lemma 2.1 and 2.2, we have
$$\begin{aligned}
&(\Delta_p^t + \Delta_d^t) - (\Delta_p^{t-1} + \Delta_d^{t-1})\\
&= (\Delta_p^t - \Delta_p^{t-1}) + (\Delta_d^t - \Delta_d^{t-1})\\
&\leq -\rho\left(\|\Delta w_k^t\|^2 + \|\Delta z_k^t\|^2\right)\\
&\quad + \eta\left(\|w_k^t - z^t\|^2 - 2(w_k^t - z^t)^T(\bar{w}_k^t - \bar{z}^t)\right).
\end{aligned}$$

Taking expectation on both sides w.r.t. the random selected indexes $k_1, k_2, ..., k_R$, we have
$$\begin{aligned}
&E[\Delta_p^t + \Delta_d^t] - (\Delta_p^{t-1} + \Delta_d^{t-1})\\
&\leq -\frac{\rho R}{K}\left(\|\Delta w^t\|^2 + \|\Delta z^t\|^2\right)\\
&\quad + \frac{\eta R}{K}\left(\sum_{k=1}^{K}\|w_k^t - z^t\|^2 - 2\sum_{k=1}^{K}(w_k^t - z^t)^T(\bar{w}_k^t - \bar{z}^t)\right),
\end{aligned}$$

where $\Delta w^t$ and $\Delta z^t$ are the primal iterate of standard ADMM, and
$$\begin{aligned}
&\sum_{k=1}^{K}\|w_k^t - z^t\|^2 - 2(w_k^t - z^t)^T(\bar{w}_k^t - \bar{z}^t)\\
&= \sum_{k=1}^{K}\|(w_k^t - z^t) - (\bar{w}_k^t - \bar{z}^t)\|^2 - \|\bar{w}_k^t - \bar{z}^t\|^2\\
&\leq 2\sum_{k=1}^{K}\left(\|w_k^t - \bar{w}_k^t\|^2 + \|z^t - \bar{z}^t\|^2\right) - \|\bar{w}_k^t - \bar{z}^t\|^2.
\end{aligned}$$

Now we invoke the error bound in [5, Lemma 2.3, 2.5] to bound the distance between $(w^t, z^t)$ and $(\bar{w}^t, \bar{z}^t)$ in terms of progress in primal iterate $\|\Delta w^t\|^2 + \|\Delta z^t\|^2$ as
$$\sum_{k=1}^{K}\|w^t - \bar{w}^t\|^2 + \|z^t - \bar{z}^t\|^2 \leq \tau(\|\Delta w^t\|^2 + \|\Delta z^t\|^2), \tag{68}$$

where $\tau$ is a positive constant. Then we have
$$\begin{aligned}
&E[\Delta_p^t + \Delta_d^t] - (\Delta_p^{t-1} + \Delta_d^{t-1})\\
&\leq -\frac{R(\rho - 2\eta\tau)}{K}(\|\Delta w^t\|^2 + \|\Delta z^t\|^2) - \frac{R\eta}{K}\sum_{k=1}^{K}\|\bar{w}_k^t - \bar{z}^t\|^2,
\end{aligned} \tag{69}$$

which is always negative for $\eta < \rho/(2\tau)$. $\qquad\square$

Then we can have following theorem for linear convergence of BC-ADMM.

**Theorem 10** (BC-ADMM Convergence). *Consider a regularized ERM problem* (53) *of the form*
$$f_i(w) = L_i(\Phi_i w) + \frac{1}{N}R(w).$$

*If the loss function $L_i(.)$ is smooth, and one of $L_i(.)$ or $R(.)$ is strongly convex, Algorithm 4 converges to the optimum of* (53) *at a linear rate, that is,*
$$E[\Delta_p^t + \Delta_d^t] \leq \frac{1}{1+\lambda}(\Delta_p^{t-1} + \Delta_d^{t-1}) \tag{70}$$

*for some constant $\lambda > 0$.*

*Proof.* To prove linear convergence, we show that the two terms in (69) can be lower bounded by the current residual $\Delta_p^t, \Delta_d^t$ respectively. In particular, we invoke the error bound in [5, Lemma 3.1] that shows

$$\Delta_d^t \leq \tau_2 \|\nabla d(\boldsymbol{\mu}^t)\| = \tau_2 \|\bar{\boldsymbol{w}}^t - \bar{\boldsymbol{z}}^t\|^2 \tag{71}$$

and

$$\Delta_p^t \leq \xi \left( \|\Delta \boldsymbol{w}^t\|^2 + \|\Delta \boldsymbol{z}^t\|^2 \right) \tag{72}$$

for some positive constant $\tau_2$, $\xi$ and $\forall t \geq t_0$, where (72) has combined [5, Lemma 3.1] and (68). Apply above error bounds on (69), we have

$$E[\Delta_p^t + \Delta_d^t] - (\Delta_p^{t-1} + \Delta_d^{t-1})$$

$$\leq -\frac{R(\rho - 2\eta\tau)}{K}(\|\Delta \boldsymbol{w}^t\|^2 + \|\Delta \boldsymbol{z}^t\|^2) - \frac{R\eta}{K}\sum_{k=1}^{K}\|\bar{\boldsymbol{w}}_k^t - \bar{\boldsymbol{z}}^t\|^2$$

$$\leq -\frac{R(\rho - 2\eta\tau)}{K\xi}\Delta_p^t - \frac{R\eta}{K\tau_2}\Delta_d^t$$

$$\leq -\lambda(\Delta_p^t + \Delta_d^t),$$

where $\lambda = \frac{R}{K}\min\left\{(\rho - 2\eta\tau)\xi^{-1}, \eta\tau_2^{-1}\right\} > 0$ for step size $\eta < \rho/(2\tau)$. After rearrangement we have

$$E[\Delta_p^t + \Delta_d^t] \leq \frac{1}{1+\lambda}(\Delta_p^{t-1} + \Delta_d^{t-1}), \quad t \geq t_0.$$

$\square$

[Supplementary Material 2]

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

|   3.3. Compute $\boldsymbol{\mu}^t_{B_k}$ from (8). |     3.1.2. Load $\mathcal{D}_{B_k}, \boldsymbol{\alpha}^t_{B_k}$ into memory. |
|   3.4. Solve (7) to obtain $(\boldsymbol{w}^*, \boldsymbol{\xi}^*_{B_k})$. |     3.1.3. Compute $\boldsymbol{\mu}^{(t,s)}_{B_k}$ from (15). |
|   3.5. Compute $\boldsymbol{\alpha}^{t+1}_{B_k}$ by (9). |     3.1.4. Solve (14) to obtain $(\boldsymbol{w}^*, \boldsymbol{\xi}^*_{B_k})$. |
|   3.6. Maintain $\boldsymbol{\mu}^{t+1}$ through (10). |     3.1.5. Compute $\boldsymbol{\alpha}^{t+1}_{B_k}$ by (16). |
|   3.7. Save $\boldsymbol{\alpha}^{t+1}_{B_k}$ out of memory. |     3.1.6. Maintain $\boldsymbol{\mu}^{t+1}$ through (17). |