[Reviews · NeurIPS 2015]

Submitted by Assigned_Reviewer_1

Summary -------- The paper addresses convex large scale empirical risk minimization (ERM) problem with the objective function composed of the linear combination of a data-fitting term and a regularizer. Specifically the paper considers a limited memory context and proposes a block coordinate strategy. First a dual block coordinate (Dual-BCD) minimization is explored. However its global convergence requires the regularizer to be strongly convex which excludes several common regularization terms as sparsity inducing regularizers. To circumvent the issue, a Dual-Augmented Block Coordinate minimization is considered and consists at each iteration to add a proximity term to the ERM objective function and applies Dual-BCD. The Dual-Augmented approach inspires from DAL (Dual Augmented Lagrangian) method. The proximity term allows to achieve the strong convexity assumption and hence the global convergence. The convergence properties in objective function (required to be strictly convex) of the resulting algorithm is analyzed. Empirical evaluations are finally conducted to study the effectiveness of the methods.

Comments --------- * The presentation of the paper is fairly easy to follow and the derivations of Dual-BCD algorithm and Dual-Augmented-BCD (DA-BCD) methods are clearly stated. The convergence properties and the rate of convergence of the latter algorithm are provided. Empirical evaluations demonstrated the

good performances in terms of convergence and generalization error of DA-BCD. As such the material of the paper covers the classical range of optimization-oriented machine learning paper: derivation of an algorithm, study of convergence properties and comparison with existing methods.

* Theorem 3 provides convergence in objective function. What about convergence in iterates ${w^t}$. Does the sequence ${\| w^* - w^t\|}$ share the same convergence rate as the objective function ?

* In section 4.3, it was mentioned that inexact minimization can be carried out in practice without hinding the fast convergence. How this observation translates into theoretical analysis of convergence ? In other words how looks the convergence theorem when inexact minimization is performed ?

* Implementation of DA-BCD algorithm involves the choice of the step-sizes $\eta_t$. How these parameters can be chosen along the iterations of the algorithm ?

Summary: The paper covers the classical range of optimization-oriented machine learning paper: derivation of an algorithm, study of convergence properties and comparison with existing methods. The obtained results are convincing.

Submitted by Assigned_Reviewer_2

The main result is the linear convergence rate in terms of outer rounds of the method, assuming exact solvers of the (still large) ERM subproblems which fit in memory.

The presentation of the paper could be improved significantly, but I think the contributions are still valuable, practical and interesting. Several of my comments below are suggestions for further improvements. UPDATE: Unfortunately the experiments lack a comparison to Quasi-Newton methods and d-GLMNET (i.e. the serialized variants of these). this issue was not answered in the author feedback, which questions parts of the value of the experiments.

1) Exact solution of the subproblems is assumed (See also Section 4.3.1), which is disappointing in view of practical applications. Even Theorem 2 only provides approximate solutions for the inner solvers. On the positive side, the experiments already work with approximate inner solvers.

2) Choice of \eta_t in theory and experiments: In the paper, please make more clear that the algorithm depends on this step-size parameter. In experiments, how was this parameter selected? How does the performance depend on the parameter?

3) Basic consensus-type ADMM variants which are well-studied for this type of problem. Their discussion (i.e. the summary) should be moved from the appendix to the main paper, and made more precise. Currently a good amount of space in the main paper is used by redundant formulas (e.g. useless dual, if working on the augmented primal, many repetitions from Section 3 in Section 4). It would be important to more clearly explain the difference in the updates proposed here as to the ADMM updates currently in the Appendix (Algorithm 3 and 4). I.e. write the actual ADMM updates and L_k for the case here, and compare to (15).

4) Experiments:

The paper unfortunately lacks an experimental comparison with other state of the art distributed l1-solvers (i.e. their serialized counterparts), such as d-GLMNET [A], Shotgun [B] and most importantly orthant-wise variants of Quasi-Newton methods [C] which are standard for l1-problems.

The experiments are currently presented in a way that is far from reproducible, and we'd hope this will be improved. For example, the stopping criteria for the subproblem solvers, the \eta_k parameters, dependence on the regularization parameters etc is not given. Also, timing plots alone might not be very informative, but it might be important to discuss the IO load and number of passes through the data. In particular for online methods such as MD/SGD, they might suffer unfairly from inefficient random access, but could also be run in a memory-aware fashion. The tradeoff of IO vs computation could also be investigated, i.e. how much does one benefit from more local computation before swapping in a new data block.

5) The discussion around lines 422 suggests that in contrast to MD, the approach here has linear convergence. This however is very oversimplifying, as the linear convergence is in number of outer rounds, each round consisting in an exact solution of a problem in millions of datapoints (same structure as the original problem). In contrast, the convergence rates of the online methods is in terms of individual points processed. The constants in the linear rate theory should be discussed, and could potentially be bounded in some relevant cases, e.g. for Lasso.

6) The level of generality and the function classes assumed for the main optimization problems should be stated more precisly.

7) *Dual* nature of the algorithm: Why do you call the algorithm 'dual-BCD' (motivated by [31]), if it is here applied to (7), which is the primal problem, with augmentation? This also relates to 'dual' in the title of the paper, as well as the title of Section 3, which in this view is confusing to start with the dual (4) if in the end the interest is (7). The reader will always be confused since all your proposed 'dual' methods are not solving (4) but (7). Also in line 247, calling it just BCD (or primal BCD or saddle-point BCD) would make it easier for the reader to understand which methods work on the dual, and which on the primal.

8) Presentation The paper in some places seems to have been written in a rush, and the presentation and related work discussions could be greatly improved. In particular, the paper would benefit from a more concise discussion of related proximal gradient methods and their rates, as there is existing literature already even for the case of approximate proximal steps.

9) Why don't you generalize your result by allowing arbitrary solvers for the subproblem, why restricting to BCD (Theorem 2 etc), especially as this gets easily confused with outer BCD? In this respect, Theorem 3 is crucial for your contribution, while Theorem 2 is not.

10) Minor: In Section 3, the authors extend the block-minimization framework of [31,2] from linear SVM to general ERM problems. However, this was already done by [D,E] for the ERM dual, though not including the l1 primal case. The transformation of those algorithms from the distributed setting to the 'limited memory' case here is straightforward, same as for ADMM and related methods.

11) About easily adapting local solvers around line 280: Make precise the comment only covers first-order solvers, not solvers which potentially rely on more problem structure.

12) Clarify how \mu_Bk is a variable in (7), i.e. if alpha_Bk is also an optimization variable. Same later in (15) etc.

minor - w^*() in line 3.2 of algorithm 2 is not clearly defined. -the readability of the whole paper and in particular the algorithms could easily be improved. 064: descnet 065,321: subsumes -> includes 176: define the function F, \bar F (objective in which problem?) 286: _in is not defined and properly related with S or Theorem 2. appendix: 581: appendix of lemma? 589: prposed. 2-norm messed up 630: define assumptions on \mu in Theorem 5 695: w and z are vectors in (38), not numbers, so need norms 784: note that 785: the analysis of

*** additional references:

[A] I Trofimov and A Genkin. Distributed Coordinate Descent for L1-regularized Logistic Regression. arXiv.org, 2014.

[B] J K Bradley, A Kyrola, D Bickson, and C Guestrin. Parallel Coordinate Descent for L1-Regularized Loss Minimization. ICML, 2011.

[C] G Andrew and J Gao. Scalable training of L1-regularized log-linear models. ICML, 2007.

[D] T Yang. Trading Computation for Communication: Distributed Stochastic Dual Coordinate Ascent. NIPS, 2013.

[E] M Jaggi et al. Communication-Efficient Distributed Dual Coordinate Ascent. NIPS, 2014.
Summary: The paper studies limited memory algorithms for general ERM problems, a highly relevant problem, and gives a valuable, practical new approach in an increased level of generality. The approach starts from the standard (dual) block-coordinate descent, but modifies this by adding an l2 proximal term. The presentation of the paper should be improved.

Submitted by Assigned_Reviewer_3

This paper proposes a dual (augmented) block coordinate algorithm to solve general regularized Empirical Risk Minimization (ERM) problems when data cannot fit into memory. Theoretical analysis is presented to show that the algorithm converges globally. Empirical studies are also conducted; results show that the proposed algorithm works well.

Major concerns:

I don't think the claim in footnote 1 (page 3) is correct. When the data matrix is not of full column rank, the objective function (not including the L2-reuglarizer) of the logistic regression is always not strongly convex.

It is not that obvious to me how (7) can be obtained from (6). I think this step is quite critical to the follow-up algorithm design, as it is the "decomposable step" w. r. t. the data blocks. It is quite necessary to provide more details about this (at least in the supplementary material).

In the Dual-Augmented Block Coordinate Descent (DA-BCD) algorithm, how to choose the parameter $\eta_t$ plays a critical role in the fast convergence. Although Theorem 3 points out that $\eta_t$ should be greater than some constant, it is still not clear how $\eta_t$ should be chosen in the implementation.

It would be better if the results can show how much time the proposed algorithm takes on computation and I/O, respectively.

Typo:

"In Figure 5, .....": I only found one figure in the paper.
Summary: See comments below.

Submitted by Assigned_Reviewer_4

This paper deals with the problem of designing optimization algorithms for model fitting on a single machine when the random access memory is not sufficient to hold all the data. More specifically, the paper deals with the regularized empirical risk minimization problem where the objective function is the sum of a loss function and a regularization term. This setting has been dealt with before in using algorithms such as block coordinate descent, but the paper claims that these algorithms may not converge if the objective is not smooth, which does happen in many cases of interest. The paper first adapts existing algorithms to the more general setting, then observes that this may not converge if the objective function is not smooth, and then modifies it using the Dual Augmented Lagrangian (DAL) method. There are experimental results that show that the algorithm proposed here is considerably better than the previous state of the art.

I don't understand the sentence on line 232: "According to Theorem 1, the Block Minimization Algorithm 1 is not globally convergent if R(w) is not strongly convex,.." I don't see how this follows from Theorem 1, unless you construct a specific problem and show that Algorithm 1 does not converge for that problem.

The paper does not give any references for Proximal Point Methods/DAL method. Given that this method is central to this paper, I think it's important to give one or two references and also have a very brief discussion of its past uses.

The problem dealt with here is clearly important and the experimental results seem very good. The paper is not very innovative in the sense that it's basically applying a well-known technique (DAL) which is tailor-made for the case of optimization problems where the objective function is not smooth, to a more specialized optimization domain (block minimization problem with the objective function not being smooth etc.). Also, as indicated above, it does not prove theoretically that the previous algorithms (as adapted in Algorithm 1 in the paper) do not converge. In the plots it does seem to converge albeit slowly. While this is a more or less straightforward application of DAL to the current setting, if it had not been done before then I think the results here are of value.

In summary, the paper can perhaps be accepted but mainly on the strength of the experimental results.

Writing can be improved considerably. There are also many typos.

Summary: The paper can perhaps be accepted but mainly on the strength of the experimental results.

Submitted by Assigned_Reviewer_5

(light review) l18 training linear l416 Figure 1 not 5
Summary: The method proposed is well explained and the experimental section demonstrate a significant speed-up compared to competing limited memory algorithms. The methods involved (block updates, overcoming non smoothness with augmeneted dual methods) are not new, but this is a well executed paper that tackle convicingly the memory challenge of huge data.

Author Feedback
Author rebuttal: We are thankful to all reviewers for their careful and constructive comments.

To Reviewer_1:

1. Dual nature of the algorithm. (Why called dual BCD?)

We call the algorithm dual Block Coordinate Descent because it optimizes w.r.t. one block of dual variables \alpha_B at a time. What might be confusing is that we update \alpha_B by solving (7) (or augmented version (15)), which is the dual of dual block minimization problem.

2. Why not generalize to arbitrary solver but restricted to BCD ?

We did allow arbitrary solver to be used for the block sub-problem (7) (or its augmented version (15)). The dual BCD is only used to decompose problem into block sub-problems, namely (7), (15).

3. Choice of \eta_t, stopping criteria and regularization parameter.

In our experiments, we set \eta_t=1 and \lambda=1 for all data sets, and two epoches of block minimization are run for each proximal update (12). We will add those explanation into experimental section.

Typically a larger \eta_t makes more conservative proximal updates but allows faster convergence of the inner sub-problem. Therefore, a larger \eta_t increases the #outer iterations required but reduce the #inner iterations required.

4. Inexact solution of sub-problems.

The analysis of BCD [20] allows inexact minimization (even for only one step of proximal gradient on sub-problem (15)).

For the Proximal-Point iteration (12), we came up with analysis to relax the requirement of exactness for each proximal update after submission. In particular, suppose each proximal sub-problem (12) is only solved up to \eps_0 precision s.t. the inexact \hat{w}^{t+1} satifies

\| \hat{w}^{t+1}-\prox( \hat{w}^t ) \| \leq \eps_0

then by non-expansiveness of proximal mapping, we can show that

\|\hat{w}^{t+1} - w^{t+1}\|

= \|\hat{w}^{t+1} - \prox(\hat{w}^{t})\| + \|\prox(\hat{w}^{t})-\prox(w^{t})\|

\leq \eps_0 + \|\hat{w}^t - w^t\|,

where w^{t} denotes the sequence of exact updates. Then for T outer iterations we will lose at most T\eps_0 additional precision. Therefore, solving each proximal problem (12) to precision \frac{\delta}{T} guarantees the final precision to be better than 2\delta.

We will add analysis for the inexact subproblem into write-up or appendix.

5. I/O in MD/SGD.

In MD/SGD, we did run in a memory-aware fashion. In particular, we shuffle data first and read block by block after performing one block of SGD updates. The MD/SGD algorithm is not I/O-efficient by itself since they can hardly re-use data already in memory.

6. Relation to the general ERM formulation in distributed learning papers [D,E].

The ERM formulation in [D,E} assumes strong convex induced by L2-regularizer, which allows them to derive a smooth dual objective. In addition, the loss formulation in [D,E] does not include multi-class, ranking loss, and other loss for structured prediction. Our formulation is more general in that it allows arbitrary convex regularizers and also structured loss function.

[D] T Yang. Trading Computation for Communication: distributed Stochastic Dual Coordinate Ascent. NIPS 2013.

[E] M Maggi et al. Communication-efficient Distributed Dual Coordinate Ascent. NIPS, 2014.

(We appreciate other comments and will improve write-up based on those)

To Reviewr_2:

1. Strong convexity of logistic loss (in footnote 1).

We agree that L(\Phi w) is not strongly convex w.r.t. w due to possible rank deficiency of \Phi.

What we mean in the footnote is that L(\xi) is strongly convex w.r.t. \xi (instead of w) for \xi in a bounded region.

2. Choice of \eta_t.

See 3rd point to Reviewer_1.

(We appreciate other comments and will improve write-up based on those)

To Reviewer_3:

1. Why is (non-augmented) dual BCD not globally convergent?

What we mean is that, according to Theorem 1, the Block Minimization Algorithm 1 is not "guaranteed" convergent to global optimum. We will state this in a clearer way.

In experiment of L1-regularized problem, the (non-augmented) dual BCD did stuck at a much higher objective than that of other globally convergent solvers.

2. Missing reference on proximal-point method and DAL.

[4,25] are our primary references to proximal-point and DAL. However, we will try to add more discussion on this thread of research.

(We appreciate other comments and will improve write-up based on those)

To Reviewer_5:

1. Convergence of solution \|w^t-w^*\| instead of function value?

The proximal-point algorithm did converge in terms of \|w^t-w^*_t\|, where w^*_t denotes the projection of w^t to optimal solution set, as shown in [25] of our reference. We will try to include a corollary that point this out.

2. Convergence under inexact minimization of sub-problem?

See 4th point to Reviewer_1.

3. Choice of \eta_t.

See 3rd point to Reviewer_1.